# PROBABILISTIC AUDITS FOR VERIFIABLE TRAINING AND OUTCOME IMPROVEMENT IN DECENTRALIZED LEARNING

## ABSTRACT

Decentralized training of large models presents two critical verification challenges: ensuring the training process was executed correctly (process verification) and confirming the resulting model genuinely improved (outcome verification). Existing solutions like zkML are prohibitively expensive, while prior Proof-of-Learning schemes focus only on the process, failing to guarantee that the final model is actually better. We introduce a comprehensive and efficient framework that addresses both challenges through economically-secured probabilistic audits. First, we propose a protocol where Provers commit to each training step, with a small, random fraction of steps being audited by verifier committees, and we derive a tight detection-cost frontier that minimizes verification overhead. Second, we introduce Proof-of-Improvement (PoI), a novel and lightweight evaluation audit that statistically certifies milestone-based gains (e.g., perplexity reduction) on a committed dataset. Empirically, on a QLoRA fine-tuning task, our process audits reduce verification compute by over 95% compared to full replication, and our PoI audits certify model improvements with high statistical power at a minimal cost.

## 1 INTRODUCTION

Decentralized training of large models raises a fundamental question: *how can a coordinator (or the public) verify that training was executed faithfully and in such a way that the resulting model improved on a committed evaluation?* Purely cryptographic approaches (e.g., zk-ML) offer strong guarantees but remain orders of magnitude too costly at training scale; e.g., even small training rounds exhibit heavy proof-generation overheads Lavin et al. (2024); Bellachia et al. (2025). Proof-of-Learning (PoL) replaces circuits with replayed spot-checks over a logged trajectory Jia et al. (2021), and Proof-of-Sampling (PoSP) couples random verification with penalties Zhang & Wang (2024); Zhao et al. (2024). However, when applied to *decentralized*, *trustless*, LLM-scale training, these approaches often require redundancy in their verification (in order to avoid adversarial effects from the *auditor* side), and hence, either inherit the cost of frequent replays or force delicate incentive/committee trade-offs that can create high entry barriers.

In this work, we focus on *training-time audits that are both economical and ML-relevant*. Building on the observation that auditing a random fraction of steps suffices to achieve a target single-step detection level, we analyze and instantiate a protocol with (i) binding per-step commitments, (ii) *post-commit* randomized audits by small verifier committees, and (iii) explicit incentives. A key outcome is a simple, actionable *detection–cost frontier*: the probability of catching a single forged step, $\delta(1)$, scales linearly with the audited fraction, $\delta(1) = \alpha q$, where $q$ captures committee capture and numerical tolerance; the minimal verification cost for a target $\delta^\star$ then follows in closed form. A windowed commit→sample→reveal→audit pipeline ensures $O(\ell)$ liveness with only a constant per-window overhead, and a tight stake bound enforces incentive compatibility. We also extend the mechanism to a multi-trainer regime (DiLoCo/Streaming-DiLoCo style, cf. Douillard et al. (2023; 2025)) where outer-round aggregation is audited and, on failure, sampled inner audits attribute blame Douillard et al. (2023; 2025). All of these pieces are validated empirically on a QLoRA fine-tuning workload (cf. Dettmers et al. (2023)), matching the predicted $\delta(1) = \alpha q$ law and delivering considerable compute savings versus fully redundant PoL.

Beyond verifying *how* training was executed, we introduce a lightweight evaluation and audit track that verifies *what* training achieved. At chosen milestones (e.g., end of a window), the trainer claims an improvement of at least $\gamma$ in token log-loss (equivalently, perplexity) over a committed baseline and evaluation root. The coordinator then (post-commit) samples $n$ evaluation tokens and assigns a small committee to recompute *forward-only* log-probs for both models; the claim is accepted if a pre-declared one-sided test (or sequential test) confirms improvement at confidence $1 - \alpha_{\text{stat}}$. The resulting detection of false improvement claims factorizes as $\delta_{\text{PoI}} = \delta_{\text{stat}}(n) \cdot q$ (statistical power times committee factor), and the verification cost is linear in both the committee size and the number of evaluated tokens, reusing the same economics and incentives as training-step audits.

## 1.1 RELATIONSHIP TO PRIOR WORK.

Zero-Knowledge Machine Learning (zkML) frameworks Lavin et al. (2024); Bellachia et al. (2025) target cryptographic correctness but face prohibitive proof costs at training scale. PoL establishes replay-based verifiability through spot-checks Jia et al. (2021), and PoSP introduces randomized committees with economic penalties Zhang & Wang (2024); Zhao et al. (2024). Our work complements these lines by: (i) deriving a *tight* detection–cost frontier with explicit committee-capture and tolerance factors; (ii) proving pipelined liveness and incentive compatibility under a realistic, committee-voted audit; (iii) extending to multi-trainer training with attribution; and (iv) adding a *Proof-of-Improvement* track that certifies outcome gains on a committed evaluation with statistical confidence. Unlike robust aggregation methods (e.g., FLTrust Cao et al. (2021)) which filter updates during training, PoI certifies the final realized improvement, securing the outcome against both lazy workers and model-poisoning attacks. On the distributed side, our multi-trainer extension is designed to coexist with low-communication training methods, such as DiLoCo and Streaming-DiLoCo Douillard et al. (2023; 2025), providing the missing verification layer. The interested reader can find a more thorough literature review in Appendix A, and a discussion of defenses against PoL vulnerabilities (e.g., Fang et al. (2023)) in Appendix A.4.

## 1.2 CONTRIBUTIONS.

We present several mechanisms for auditing the veracity and effectiveness of training and fine-tuning in the context of a decentralized and trustless environment. Specifically, our main contributions are:

- **Protocol.** A practical commit→sample→reveal→audit protocol with small verifier committees and binding per-step commitments (inputs, outputs, metadata). This is done on a windowed schedule, which enables concurrency.
- **Detection–cost frontier.** We derive theoretical results for the detection probability, and give the *minimal* verification cost for any target $\delta^\star$, revealing a simple linear frontier that replaces $M$ full replays with $m \ll M$ audited replays.
- **Liveness and incentives.** We bound end-to-end wall-time under windowed auditing and give a tight stake condition that makes honesty strictly dominant; a repeated-interaction variant further reduces collateral.
- **Multi-trainer extension.** An outer-round aggregation audit with sampled inner audits on failure attributes faults to workers, preserving low typical-case cost while enabling slashing at worker granularity.
- **Proof-of-Improvement (PoI).** A drop-in evaluation-audit track that certifies milestone improvements (e.g., perplexity decrease by $\geq \gamma$) on a committed evaluation set; $\delta_{\text{PoI}} = \delta_{\text{stat}}(n) \cdot q$ and verification cost is linear in committee size and audited tokens.

The rest of this work is organized as follows. We present the core methodologies in Section 2. We devote Section 3 to the analysis (with proofs in Appendix C), and experimentally validate our research in Section 4. Lastly, we present some closing remarks and limitations in Section 5.

## 2 METHOD: PROBABILISTIC AUDITS WITH COMMITTEES

In the distributed variant, we run *background* worker audits every window: a $\beta$ fraction of workers are sampled (via public randomness and a VRF) and, within each sampled worker, an $\alpha$ fraction of steps are replayed by a committee of size $m$ under tolerance $\tau$. If either the aggregation check or the

PoI check fails, we *escalate* auditing on the preceding window by increasing $(\alpha, \beta, m)$ (optionally to a full audit with $\beta=1$) to attribute and slash.

## 2.1 PROTOCOL PRELIMINARIES

Let $\mathcal{P}$ denote the set of protocol participants, consisting of $M_p \in \mathbb{N}_+$ *Provers*, $\{P_1, \ldots, P_{M_p}\}$, and a population of $M_v \in \mathbb{N}_+$ *Verifiers*, $\{V_1, \ldots, V_{M_v}\}$. A given Prover $P$ performs $\ell \in \mathbb{N}_+$ gradient-update steps on model parameters $\theta \in \mathbb{R}^d$. Each step $t$ transforms $\theta_{t-1}$ into $\theta_t$ via a (possibly randomized) update function:

$$\theta_t = \mathsf{Update}(\theta_{t-1}, \mathcal{D}_{t-1}, \mathcal{L}_{t-1}, \mathcal{H}_{t-1}), \qquad t = 1, \ldots, \ell, \tag{1}$$

where $\mathcal{D}_{t-1}$ represents the data (e.g., indices with Merkle proofs), $\mathcal{L}_{t-1}$ is the loss function, and $\mathcal{H}_{t-1}$ contains auxiliary state such as optimizer state, RNG seeds, and environment identifiers. The Prover's per-step compute cost is $C_p > 0$, while a Verifier incurs a cost of $C_v > 0$ to replay a single step.

We assume a trusted *smart contract* ($\mathsf{SC}$) that manages participant stakes ($s_p$ and $s_v$), provides public randomness (cf. Syta et al. (2017); Choi et al. (2023), for example) , and executes the protocol logic. All authenticated messages are delivered within a known delay bound $\Delta$. We consider a *static* Byzantine adversary $\mathcal{A}$ who, prior to execution, may corrupt up to $f_p$ of the Prover's update rounds and up to $f_v$ Verifiers within any given audit subcommittee of size $m$.

Training is organized into windows of $G$ steps. In each window we use two sampling rates: a *worker rate* $\beta \in (0, 1]$ for background selection of workers to audit, and a *per-worker step rate* $\alpha \in (0, 1]$ for selecting steps within a sampled worker. Committees have size $m$ and vote under a numerical tolerance $\tau$ calibrated on the declared stack $\Xi$; we denote the resulting committee correctness by $q = q(m, \tau)$. We impose reveal and vote deadlines $(\Delta_{\text{reveal}}, \Delta_{\text{vote}})$ to ensure liveness. Window randomness is drawn *after* all per-step commitments in that window are finalized; worker and step draws use a VRF seeded by this public randomness. To avoid repeatedly skipping the same workers/steps over short horizons, we use sampling *without replacement* within the window (and over a rolling cycle for workers).

**Remark.** *While our primary analysis considers a static adversary, the post-commitment reveal structure provides inherent resilience against certain adaptive strategies. Since committees are selected using public randomness after commitments are locked in, an adversary has a limited window to corrupt the specific verifiers chosen for an audit adaptively.*

We distinguish *rational* adversaries (profit-seeking, deterred by background audits and slashing) from *malicious* adversaries (model-degrading, detected by PoI and handled by escalation). With background sampling, the per-window detection for a worker forging $f$ steps is

$$\delta_{\text{bg}}(f) = \beta \left( 1 - (1 - \alpha q)^f \right). \tag{2}$$

To ensure each worker is audited at least once every $K$ windows with miss-probability $\leq \varepsilon$, choose

$$\beta \geq 1 - \varepsilon^{1/K}. \tag{3}$$

Setting stake/slashing $S_\$$ (or increasing $\alpha, \beta$) so that $(1 - \delta_{\text{bg}})G_\$ - \delta_{\text{bg}}S_\$ < 0$ makes rational cheating unprofitable.

## 2.2 CORE PROTOCOL: PROBABILISTIC TRAINING AUDITS

The core mechanism verifies the integrity of the training process.

This algorithm is presented as pseudocode 1 in Appendix B and illustrated in Figure 3 in Appendix . After freezing the window's per-step commitments, the contract draws a public seed $r_W$ and uses a VRF (with domain separation) to sample the audited steps at rate $\alpha$; sampling is without replacement within the window. Reveals are delivered to committees (not publicly) before the reveal deadline $\Delta_{\text{reveal}}$. Witnesses (parameter shards, optimizer shards, metadata, seeds) for audited steps are disclosed to the committee over sealed channels; public artifacts remain the per-step commitments and Merkle proofs. This limits leakage while preserving verifiability.

**1. Commit Phase.** For each training step $t$ within a window, the Prover executes the update and constructs a cryptographic commitment $h_t$ that is posted to a public ledger. This commitment acts as a tamper-proof record of the claimed state transition. To ensure these commitments are both secure and scalable for large models, we employ a Merkle tree-based approach. Instead of hashing the entire parameter state, the model parameters $\theta$ are partitioned into fixed-size shards, and the Prover commits to the Merkle root of these shards. The binding commitment for each step is a constant-size hash constructed from the Merkle roots of the model state *before* ($\theta_{t-1}$) and *after* ($\theta_t$) the update, along with a *step witness* $W_t$:

$$h_t = H\left(\mathsf{MerkleRoot}(\theta_{t-1})||\mathsf{MerkleRoot}(\theta_t)||W_t\right). \tag{4}$$

The witness $W_t := \left(I_t, \Pi_t, O_{t-1}, R_t, \Lambda_t, \Xi\right)$ contains all necessary metadata to reproduce the step, such as batch indices ($I_t$), their Merkle proofs ($\Pi_t$), optimizer state ($O_{t-1}$), RNG seeds ($R_t$), hyperparameters ($\Lambda_t$), and a hash of the execution environment ($\Xi$). This structure makes the commitment binding to the specific state transition, preventing the Prover from changing their story after the fact (*ex-post* equivocation).

**2. Sample & Reveal Phase.** After the Prover has committed to all $G$ steps in a window, the smart contract uses a source of public randomness to sample a small fraction, $\alpha$, of the steps to be audited. Once these steps are chosen, the Prover is required to "open" their commitments by revealing the full step witness $W_t$ and the specific parameter shards involved in the update, along with their Merkle proofs. The post-commitment nature of the sampling is crucial; because the Prover does not know which steps will be audited until *after* they have committed to all of them, they are incentivized to perform every step honestly.

**3. Audit & Resolve Phase.** A random subcommittee of $m$ Verifiers is selected to perform the audit. For each sampled step, the Verifiers first use the revealed Merkle proofs to confirm that the provided parameter shards match the committed roots. Only then do they re-execute the training step using the witness data to produce a recomputed state $\hat{\theta}_t$. They vote to accept the step if their result is sufficiently close to the Prover's claimed result, i.e., $\|\hat{\theta}_t - \theta_t\|_{\mathsf{X}} \le \tau$, where $\tau$ is a small tolerance to account for benign numerical drift across different hardware and $\|\cdot\|_{\mathsf{X}}$ is some appropriate norm. If a supermajority of the committee rejects any step, the Prover's stake is slashed; otherwise, honest Verifiers are rewarded, and the protocol proceeds. Committees decide by supermajority ($\ge \lceil m/2 \rceil + 1$) under the fixed tolerance $\tau$; replays run in parallel to minimize wall-clock overhead.

## 2.3 EXTENSION TO MULTI-PROVER DISTRIBUTED TRAINING

Our protocol naturally extends to communication-efficient distributed settings like DiLoCo Douillard et al. (2023), where multiple workers (Provers) train in parallel with infrequent synchronization. The process operates in two distinct loops: an *inner loop* of local training and an *outer loop* for global aggregation.

In each outer round $r$, $N_m$ workers start from a common global model state $\theta_{r-1}$. Each worker $i$ then independently performs $k$ local training steps using its own data:

$$\theta_{r,j}^{(i)} = \mathsf{Update}\bigl(\theta_{r,j-1}^{(i)}, \mathcal{D}_{r,j-1}^{(i)}, \mathcal{L}_{r,j-1}^{(i)}, \mathcal{H}_{r,j-1}^{(i)}\bigr), \qquad t = 1, \ldots, \ell, \tag{5}$$

where $\theta_{r,0}^{(i)} = \theta_{r-1}$. During this phase, each worker acts as a single Prover, creating and posting commitments ($h_{r,j}^{(i)}$) for each of its $k$ local steps, just as described in Section 2.2.

After completing their $k$ local steps, each worker proposes their final local model, $\theta_{r,k}^{(i)}$. The new global model, $\theta_r$, is then computed by aggregating these proposals, for example, through simple averaging: $\theta_r = \frac{1}{N_m} \sum_i \theta_{r,k}^{(i)}$.[1]

Verification proceeds in a two-stage, optimistic fashion to minimize cost:

---

[1]In more advanced schemes like DiLoCo, this aggregation can also incorporate momentum (e.g., a Nesterov step, cf. Lin et al. (2019))

1. **Stage 1: Aggregation Audit.** First, a verifier committee performs a lightweight check on the outer loop. It verifies that the final global model $\theta_r$ was correctly computed from the workers' proposed final models $\theta_{r,k}^{(i)}$. This step is computationally cheap as it only involves re-calculating the aggregation, not re-playing any training steps.

2. **Stage 2: Fault Attribution (Escalation).** *If and only if* the aggregation audit fails, the protocol escalates. A random subset of workers $Q \subseteq \{1, \ldots, N_m\}$ is challenged to reveal their full $k$-step local training histories. The core probabilistic audit from Section 2.2 is then performed on each challenged worker to find who produced a faulty local model.

This approach is illustrated in Figure 4 in Appendix F and in Algorithm 2 in Appendix B.

**Remark** (On Background audits vs. escalation) *In addition to the optimistic two-stage flow above, we* always *run background worker audits at rate $\beta$ every window (VRF-drawn after commits freeze); within each sampled worker, an $\alpha$ fraction of steps is replayed by a committee under tolerance $\tau$. Stage 2b escalation (raising $(\alpha, \beta, m)$, optionally $\beta=1$) is triggered when Stage 1 fails or when PoI fails at a milestone; it serves to attribute faults and slash.*

### 2.4 EVALUATION AUDIT: PROOF-OF-IMPROVEMENT (PoI)

To certify that training achieved a meaningful outcome, we introduce PoI, an audit that verifies performance gains at milestones.

At the end of a window, a Prover posts a claim $(r, \gamma, \alpha_{\text{stat}})$:

> "At milestone $r$, the final model $\theta_{\text{final}}$ improves token log-loss by at least $\gamma > 0$ (reduces perplexity by a factor $\leq e^{-\gamma}$) versus baseline $\theta_0$ on the committed evaluation set, with one-sided confidence $1 - \alpha_{\text{stat}}$."

This requires a one-time, pre-run commitment to $H(\theta_0)$ and the evaluation data's Merkle root, $R_{\text{eval}}$.

The contract samples a subset $\tilde{S}_r \subseteq D_{\text{eval}}$ of size $n$. A verifier committee of size $m_{\text{eval}}$ computes per-token log-loss differences for $i \in \tilde{S}_r$:

$$Z_i := \underbrace{-\log p_{\theta_{\text{final}}}(x_i)}_{\ell_1(x_i)} - \Big(\underbrace{-\log p_{\theta_0}(x_i)}_{\ell_0(x_i)}\Big),$$

The claim of improvement is accepted if and only if a pre-declared one-sided statistical test on the sample mean $\bar{Z}$ passes. For example, by confirming that the Lower Confidence Bound (LCB) of the mean improvement meets the claimed threshold: $\text{LCB}_{1-\alpha_{\text{stat}}}\big(-\bar{Z}\big) \geq \gamma$. This mechanism, pipelined alongside training audits, provides a low-cost, statistically robust method to verify tangible model improvements without requiring monotonic per-step loss. If PoI does not certify improvement at a milestone, we trigger Stage 2b escalation (cf. Algorithm 2, raising $(\alpha, \beta, m)$; optionally $\beta=1$) for the implicated window(s) to attribute faults and apply slashing.

## 3 ANALYSIS

We present all our proofs in the Appendix C.

**Definition 1** (Soundness). *Let $\varepsilon_{\text{sound}} \in [0, 1]$. A protocol is $\varepsilon_{\text{sound}}$-sound if, against any adversary corrupting up to $f_p$ Prover-rounds and up to $f_v$ verifiers per subcommittee, the probability that any incorrect update is* not *detected is at most $\varepsilon_{\text{sound}}$.*

**Definition 2** (Liveness). *A protocol satisfies* liveness *if, when all participants are honest, all $\ell$ training rounds are completed within a total time of $T \leq \ell T_{\text{upd}} + O(\ell/\mathcal{G})$, where $T_{\text{upd}}$ is the per-step computation time and $\mathcal{G}$ is the window size.*

**Definition 3** (Incentive Compatibility). *Let $\mathcal{G} > 0$ be the Prover's expected gain from a successful one-step cheat, and let $s_p > 0$ be the at-risk stake. The protocol is* incentive-compatible *if the expected utility for cheating is negative, i.e., $U_{\text{cheat}}(f) < U_{\text{honest}} = 0$ for any number of fraudulent steps $f \geq 1$.*

## 3.1 SOUNDNESS AND COST OF PROCESS AUDITS

The soundness of our protocol hinges on the probability that a fraudulent step is both sampled for an audit and correctly flagged by an honest-majority committee.

Across windows, choosing $\beta$ per Eq. equation 3 (*e.g.*, $\beta \geq 1 - \varepsilon^{1/K}$) bounds the probability a worker evades audit for $K$ consecutive windows by $\varepsilon$. For any single audited step, the probability $q$ that a forgery is successfully detected is given by:

$$q \;=\; \underbrace{(1 - P_{\text{maj-Byz}}(M, F, m))}_{\text{Prob. of honest majority}} \cdot \underbrace{(1 - P_{\tau\text{-miss}})}_{\text{Prob. correct flag given honest majority}}, \tag{6}$$

Where $P_{\text{maj-Byz}}$ is the probability of a Byzantine majority in a committee of size $m$ drawn from a population with $F$ adversaries (given by the hypergeometric distribution), and $P_{\tau\text{-miss}}$ is the probability that numerical tolerance $\tau$ masks a genuine error. For deterministic computations, $\tau = 0$ and $P_{\tau\text{-miss}} = 0$. The overall detection probability depends on the number of fraudulent steps $f$. The base case for a single forgery is linear in the sampling rate $\alpha$.

**Lemma 1** (Base Law for Single Forgery). *For a single forged step ($f = 1$), the detection probability is exactly $\delta(1) = \alpha q$.*

For multiple forgeries, let $X \sim \text{Hypergeom}(\ell, f, \tilde{n})$ be the number of fraudulent steps sampled. The exact detection probability is the expectation over $X$. Specifically, the probability of detecting *at least* one of $f$ forged steps is:

$$\delta(f) \;=\; 1 - \mathbb{E}\big[(1 - q)^X\big] \;=\; 1 - \sum_{k=0}^{\min\{f,\tilde{n}\}} \frac{\binom{f}{k}\binom{\ell-f}{\tilde{n}-k}}{\binom{\ell}{\tilde{n}}} (1 - q)^k. \tag{7}$$

While exact, this expression can be tightly bounded for practical design, for instance, by $\delta(f) \geq 1 - (1 - \alpha q)^f$, which follows from the negative correlation of sampling without replacement (see the Appendix for proof).

## 3.2 BOUNDING THE IMPACT OF UNDETECTED FORGERIES

A key concern is whether an adversary can cause significant damage by forging only a few steps, which have lower detection probability. The following lemma bounds the impact of undetected forgeries.

**Lemma 2** (Few-Step Influence Bound). *Suppose the loss function $\mathcal{L}$ is $L$-smooth and training uses step size $\eta$ with optional gradient clipping at norm $B$. If an adversary perturbs a set $\mathcal{F}$ of $f$ steps with update errors $\{\Delta_t\}_{t \in \mathcal{F}}$ (where $\Delta_t := \theta_t^{\text{adv}} - \theta_t^{\text{honest}}$ is the per-step deviation), then:*

$$\|\theta_T^{\text{adv}} - \theta_T^{\text{honest}}\| \;\leq\; \sum_{t \in \mathcal{F}} c_t \|\Delta_t\|, \quad \text{where } c_t \leq (1 + \eta L)^{T-t}, \tag{8}$$

*and the loss deviation is bounded by:*

$$|\mathcal{L}(\theta_T^{\text{adv}}) - \mathcal{L}(\theta_T^{\text{honest}})| \;\leq\; \frac{L}{2}\|\theta_T^{\text{adv}} - \theta_T^{\text{honest}}\|^2. \tag{9}$$

*With gradient clipping, $\|\Delta_t\| \leq 2\eta B$ per forged step. Thus, forging $f$ steps causes parameter deviation $O(f \cdot \eta B \cdot (1 + \eta L)^T)$, which is bounded for typical $\eta L \ll 1$.*

The proof appears in Appendix C. This lemma shows that *even if a few forgeries escape detection,* their cumulative impact is bounded. Combined with the detection probability $\delta(f) \geq 1 - (1 - \alpha q)^f$, which grows with $f$, adversaries face a fundamental trade-off: forging many steps increases both impact and detection risk, while forging few steps limits achievable damage.

## 3.3 THE COST-SOUNDNESS FRONTIER.

The total expected computational cost is the sum of the Prover's training cost and the expected verification cost.

$$\text{Cost}_{\text{total}}(\alpha, m) \;=\; \ell C_p + \alpha \ell m C_v. \tag{10}$$

By combining Lemma 1 with the cost model, we arrive at the efficient frontier, which defines the minimum cost to achieve a target soundness level $\delta^*$.

**Theorem 1** (Efficient Frontier for Process Audits). *For a target single-step detection probability $\delta^* \in (0, q]$, the minimum achievable cost is:*

$$\text{Cost}_{\min}(\delta^*; m) \;=\; \ell C_p \;+\; \frac{\delta^*}{q}\, \ell m C_v.$$ (11)

*The equation above establishes a linear tradeoff between verification cost and soundness. Targets where $\delta^* > q$ are infeasible without improving $q$ (e.g., by increasing committee size $m$).*

Improving $q$ by sizing $m$ (and calibrating $\tau$) reduces the factor $\delta^*/q$ and thus verifier cost.

### 3.4 SOUNDNESS AND COST OF OUTCOME AUDITS (POI)

Recall that the goal here is to provide verifiable claims on improvements, rather than on computational work. To that end, notice that the detection probability for a false PoI claim is the product of the statistical power of the test and the committee quality factor:

$$\delta_{\text{PoI}} \;=\; \delta_{\text{stat}}(n) \cdot q_{\text{eval}}.$$ (12)

The cost is determined by the number of samples $n$ and the committee size $m_{\text{eval}}$:

$$\text{Cost}_{\text{PoI}}(n, m_{\text{eval}}) \;\approx\; m_{\text{eval}}\, n\big(C_{\text{eval}}(\theta_{\text{final}}) + C_{\text{eval}}(\theta_0)\big).$$ (13)

This creates a similar linear cost-soundness tradeoff, where the number of samples $n$ plays a role analogous to the sampling fraction $\alpha\ell$ in process audits. The required sample size $n$ to achieve a desired statistical power can be determined using standard results. For instance, for i.i.d. sub-Gaussian log-loss differences, $n$ scales as $n \gtrsim (\sigma/r)^2$, where $r$ is the margin of the false claim and $\sigma^2$ is the variance. When $\{Z_i\}$ exhibit topical/temporal correlation, we run blocked/paired tests (or block bootstrap) on VRF-sampled evaluation blocks; $\delta_{\text{stat}}(n)$ is then computed under the effective sample size of the block design (details in the appendix).

### 3.5 PIPELINED LIVENESS AND ECONOMIC SECURITY

Finally, we analyze the protocol's operational guarantees.

**Theorem 2** (Pipelined Liveness). *With a window size of $\mathcal{G}$, the total execution time for $\ell$ steps is bounded by:*

$$T_{\text{total}} \;\leq\; \ell\, T_{\text{upd}} \;+\; \left\lceil \frac{\ell}{\mathcal{G}} \right\rceil (2\Delta + \Delta_{\text{aud}}) \;+\; O(1),$$ (14)

*where $T_{\text{upd}}$ is the per-step update time, $\Delta$ is the network delay, and $\Delta_{\text{aud}}$ is the audit finalization time. The pipelined design ensures audit latency contributes only a constant overhead per window, not per step.*

The previous theorem demonstrates that the presented protocol achieves *liveness*, loosely speaking, meaning that the cost of the protocol increases moderately compared to the unverified case.

**Theorem 3** (Economic Security via Staking). *Honesty is a strictly dominant strategy for a Prover if their stake $s_p$ satisfies:*

$$s_p \;>\; \frac{1 - \delta(1)}{\delta(1)}\, \mathcal{G} \;=\; \frac{1 - \alpha q}{\alpha q}\, \mathcal{G},$$ (15)

*Where $\mathcal{G}$ is the gain from a single successful cheat. An analogous bound holds for PoI claims, replacing $\delta(1)$ with $\delta_{PoI}$ and $\mathcal{G}$ with the gain from a false improvement claim, $\mathcal{G}_{claim}$.*

**Corollary (with background sampling; multiple forgeries).** For a worker forging $f$ steps in a window, replace $\delta(1)$ by the background detection $\delta_{\text{bg}}(f)$ from Eq. equation 2 and replace $\mathcal{G}$ by the per-window cheating gain $\mathcal{G}(f)$ to obtain $s_p \;>\; \frac{1 - \delta_{\text{bg}}(f)}{\delta_{\text{bg}}(f)}\, \mathcal{G}(f)$. Equivalently, set deposits/rewards so that the expected utility $(1 - \delta_{\text{bg}})G_\$ - \delta_{\text{bg}} D < 0$, making rational cheating strictly unprofitable.

# 4 EXPERIMENTS

Our experiments test whether the protocol's predictions about *detection*, *cost*, and *liveness* hold in practice, and whether the outcome-audit (PoI) and the distributed (DiLoCo-style) variant behave as the analysis requires. Unless stated otherwise, we fine-tune a Phi-family Abdin et al. (2024) causal LM with LoRA/QLoRA Hu et al. (2022); Dettmers et al. (2023) adapters on WIKITEXT-2 Merity et al. (2016), and run a verifier population $M=128$ with subcommittee $m=7$. The theory in Sec. 3 predicts a single-step law $\delta(1) = \alpha q$ with $q = (1 - P_{\text{maj-Byz}})(1 - P_{\tau\text{-miss}})$ and a linear cost–soundness frontier whose intercept and slope are $1/(1+M)$ and $m/(1+M)$ when $C \simeq C_v$. We implement the windowed *commit→sample→reveal→audit* pipeline and the PoI track exactly as analyzed.

**Verifying the linear detection law** $\delta(1) = \alpha q$. We begin by testing the fundamental prediction that single-step detection scales linearly with the audited fraction. We plant a single forged update at a uniformly random step, draw an independent $m$-committee for each audited step, and sweep $\alpha \in \{0.05, \dots, 1.0\}$. The empirical detection curve is a line through the origin whose slope matches the exact $q$ from the hypergeometric committee model, confirming $\delta(1) = \alpha q$ to within binomial uncertainty (Fig. 1 (Left)). This is the base case established in the analysis and underpins all subsequent experiments. We ensure VRF draws occur post-commit and re-sample committees per step; CIs are Clopper–Pearson binomial intervals.

**Impact of numerical tolerance** Next we isolate how the numeric tolerance $\tau$—used to absorb benign cross-hardware drift—affects the committee factor $q$. We vary $\tau$, estimate the induced $P_{\tau\text{-miss}}$, and plot $q = (1 - P_{\text{maj-Byz}})(1 - P_{\tau\text{-miss}})$ (Fig. 1 (Middle)). As $\tau$ crosses a few multiples of the honest-drift scale, $q$ drops sharply; geometrically, the $\delta(1)$ line rotates downward exactly as predicted by the model. In our deterministic baseline we set $\tau=0$ (hence $P_{\tau\text{-miss}}=0$); for heterogeneous deployments we calibrate $\tau$ to a high percentile of observed honest drift on the declared stack $\Xi$. Table 3 provides sizing guidance: for a target $q \geq 0.95$ at capture fraction $F/M = 0.10$, committee size $m = 3$ suffices. Calibration follows the p99 honest replay drift under $\Xi$; cross-hardware calibration tables and numeric-error models are reported in the Appendix.

**Process-audit cost frontier** We then examine the verification cost **relative to fully redundant PoL**. Measured normalized cost aligns with the linear frontier that replaces $M$ full replays with $m \ll M$ audited replays: intercept $1/(1+M)$ and slope $m/(1+M)$ when $C \simeq C_v$. Representative operating points lie on the predicted line to plotting precision: for targets $\delta^* = \{0.50, 0.80, 0.95\}$ we achieve $\alpha \approx \{0.501, 0.802, 0.952\}$ at normalized costs $\{3.49\%, 5.12\%, 5.94\%\}$ of PoL (Table 2). These values are the ones used later when we compare to wall-clock measurements. Measured network egress per audited step matches the bytes model (*params + optimizer + proofs*); LoRA/QLoRA reduces the updated fraction $\kappa \ll 1$, cutting reveal size and masking network latency in the pipeline (details in the Appendix).

**Outcome verification with Proof-of-Improvement** We also verify *what* training achieved. At a milestone, the prover claims an improvement $\geq \gamma$ in token log-loss versus a committed baseline; the contract samples $n$ spans and a small evaluation committee runs a one-sided test. On WIKITEXT-2 with a Phi-family LM (LoRA), the full evaluation over 800 spans reports $\Delta_{\text{full}} = 0.4171$ nats/token; sampled audits with $n \in \{50, 100, 200, 400\}$ reject $H_0: \mu \leq 0$ with very high confidence (Table 4; e.g., $p = 2.76 \times 10^{-14}$ at $n=50$). This behavior matches the factorization $\delta_{\text{PoI}} = \delta_{\text{stat}}(n) \cdot q$ and the linear verification cost in $n\, m_{\text{eval}}$ from the analysis. We draw VRF-sampled *blocks* from the committed evaluation root and use paired tests to preserve power under topical correlation; the observed rejection rates align with the blocked-power predictions in the analysis.

**Cost validation on the Phi family (measured vs. theory)** To validate the cost model numerically, we measure prover step time $C$ and verifier replay time $C_v$ on the same Phi-family workload in deterministic mode ($F=0 \Rightarrow q=1$). The logs show $C_{\text{mean}} = 0.2231$s (median 0.2148s; p95 0.2278s) and $C_{v,\text{mean}} = 0.2149$s (median 0.2134s; p95 0.2243s), confirming the regime $C \simeq C_v$ assumed by the frontier. Using these times with $M=128, m=7$, the measured normalized cost tracks the theoretical line almost perfectly when plotted against $\alpha$ (Fig. 1 (Left)). The same operating points as in Table 2 are realized by the wall-clock measurements (e.g., $\delta^* = 0.80$ at $\alpha \approx 0.802$ costs $\approx 5.12\%$ of PoL), strengthening external validity of the linear law on real compute. We additionally

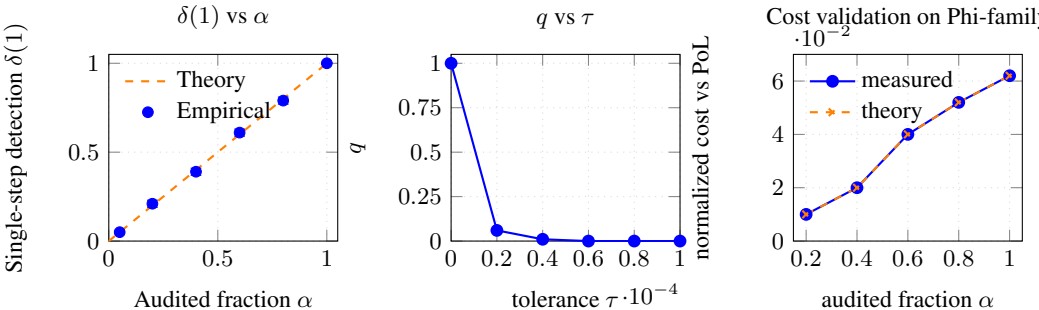

Figure 1: **(Left)** Single-step detection $\widehat{\delta}(1)$ with 95% CIs vs. $\alpha$ under committee size $m=7$ and global capture $F/M=0.10$; dashed line shows theoretical $\delta(1) = \alpha q$. **(Middle)** Committee correctness $q(= (1 - p_{\mathrm{maj} \geq m/2})(1 - P_{\tau,\mathrm{miss}}))$ as a function of tolerance $\tau$; increasing $\tau$ reduces $q$ via $P_{\tau\text{-miss}}$. **(Right)** Normalized verification cost vs. $\alpha$, confirming the linear frontier. All plots use 11pt axis labels and $1.5\times$ line weights for legibility.

confirm that wall-clock overhead scales with $\alpha$ as predicted when network transfer is overlapped (pipeline coefficient $\chi \approx 0$ in the bytes model); raw timing logs and harness configs are included in the Appendix.

**Prover overhead: commit and reveal costs**   Table 1 reports the Prover's per-step overhead, addressing the cost of Merkle commitments and witness serialization. For LoRA/QLoRA fine-tuning, the commit overhead is modest because only adapter parameters ($\kappa \ll 1$) are hashed. Reveal bandwidth per audited step follows Eq. equation 18 in Appendix D; with LoRA adapters and content-addressed serving, the network term is negligible in compute-bound regimes. Commit overhead is optimizer-agnostic; serialization depends on whether optimizer moments are included (Adam) or absent (SGD). Reveal sizes follow the scaling $B_{\mathrm{step}} \approx \kappa(1 + \phi)(P + L_p\pi)$. We remark that The prototype is intentionally naive and re-hashes all LoRA parameters on CPU each step, so its wall-clock costs are an upper bound, not representative of an optimized implementation

| Regime | Commit Overhead (% of step time) | Serialize Overhead (% of step time) | Reveal Size (MB per audited step) |
|---|---|---|---|
| LoRA + Adam | 8–12% | 8–15% | $\approx 120$ |
| LoRA + SGD | 8–12% | 1–3% | $\approx 40$ |
| Full Params + Adam | 20–30% | 15–25% | $\approx 850$ |

Table 1: Prover-side overheads for the three training regimes.

**Auditing distributed training (DiLoCo-style) with attribution**   Finally, we exercise the two-stage distributed audit: an outer-round aggregation check with sampled inner audits for attribution on failure. In a toy run with $N_m=2$ workers, $k=3$ local steps per outer round, $R=2$ outer rounds, outer audit rate $\alpha_{\mathrm{out}}=1$, inner escalation rate $\beta_{\mathrm{inner}}=0.5$, and tight tolerance $\tau = 10^{-6}$, we inject both a faulty local step and a faulty aggregation. The outer audit flags a failure and the escalation identifies a faulty worker (`identified_any_faulty=True`), demonstrating end-to-end detection and attribution under the design analyzed in Sec. 3. Background worker audits were also enabled at rate $\beta$ (even when Stage 1 passed); across repeated windows the observed single-step detection matched $\delta_{\mathrm{bg}}(1) = \beta \, \alpha q$ within binomial uncertainty, and escalation correctly attributed the faulty worker.

Across these six experiments, we find consistent agreement between practice and theory. The single-step law $\delta(1) = \alpha q$ holds; tolerance $\tau$ depresses $q$ exactly as predicted; process audits obey a linear cost frontier whose slope/intercept are confirmed by wall-clock timings; PoI delivers statistical power with verification cost linear in $n$; and the distributed variant detects aggregation failures and attributes blame with low typical-case overhead.

| $\delta^*$ | required $\alpha$ | normalized cost (%) | comment |
|---|---|---|---|
| 0.50 | $\approx 0.501$ | $\approx 3.49$ | near half detection at $\sim$3.5% of PoL |
| 0.80 | $\approx 0.802$ | $\approx 5.12$ | $\sim$95% savings vs PoL |
| 0.95 | $\approx 0.952$ | $\approx 5.94$ | $\sim$94%+ savings |

Table 2: Normalized cost vs. detection target $\delta$ (with $M{=}128$, $m{=}7$).

| $q$ | 0.05 | 0.10 | 0.20 | 0.30 | 0.40 |
|---|---|---|---|---|---|
| $\geq 99$ | 3 | 5 | 11 | — | — |
| $\geq 95$ | 3 | 3 | 7 | 15 | — |

Table 3: Minimal odd committee sizes ($F/M$ ratios abbreviated in header).

| $n$ | $\widehat{\Delta}(n)$ | $\mathrm{std}(Z)$ | $p$-val | rej $H_0$ | $\Delta_{\text{full}}$ | \|diff\| |
|---|---|---|---|---|---|---|
| 50 | 0.4650 | 0.3164 | 2.7-14 | T | | 0.048 |
| 100 | 0.5029 | 0.3779 | $<$ -16 | T | 0.417 | 0.086 |
| 200 | 0.4988 | 0.3947 | $<$ -16 | T | | 0.082 |
| 400 | 0.5027 | 0.3784 | $<$ -16 | T | | 0.086 |

Table 4: One-sided $t$-test results ($\alpha_{\text{stat}} = 0.05$).

## 5 CLOSING REMARKS

In this work we addressed two fundamental challenges in decentralized model training: ensuring the training was executed correctly (process verification) and confirming the resulting model genuinely improved (outcome verification). We introduced a comprehensive framework that combines efficient, economically secured probabilistic audits for training steps with a novel and lightweight evaluation audit we term Proof-of-Improvement (PoI). Our process audits leverage a commit-sample-reveal protocol with verifier committees to achieve high security guarantees at a fraction of the cost of exhaustive replay methods. PoI complements this by enabling provers to make statistically verifiable claims about performance gains on a committed dataset. Our theoretical analysis established a clear and actionable linear trade-off between verification cost and security, encapsulated by the single-step detection law $\delta(1) = \alpha q$ and a minimal cost frontier for achieving any target detection level. These theoretical predictions were validated empirically on a QLoRA fine-tuning task, where our protocol reduced verification compute by over 95% compared to fully redundant PoL while maintaining strong detection guarantees.

### 5.1 LIMITATIONS

Our empirical validation was focused on a fine-tuning workload, and our multi-trainer experiment was designed to demonstrate the fault-attribution mechanism rather than operate at a large scale. Extending this evaluation to more complex scenarios like large-scale pre-training is an important next step.

In addition, our analysis focuses primarily on verification compute savings and does not fully quantify the overheads incurred by the Prover. The per-step cryptographic commitment, which involves calculating a Merkle root over the model's entire state, introduces computational costs that scale with model size. Furthermore, the 'Reveal" phase for audited steps requires transmitting the step witness, which includes the optimizer state. For optimizers like Adam, this state can be substantial, representing a potential communication bottleneck that merits further investigation, especially in bandwidth-constrained decentralized environments.

Furthermore, we acknowledge that our security analysis primarily considers a static Byzantine adversary. While the post-commitment reveal structure offers some protection, the protocol's resilience against more sophisticated, adaptive adversaries warrants a deeper investigation. Such adversaries, who might attempt to corrupt verifiers after a committee is selected, are countered by the assumption of a short finalization window. Future work should formally analyze the assumptions required to secure this window and explore stronger cryptographic mechanisms to mitigate these adaptive risks.

Finally, our implementation of PoI certifies improvement based on log-loss reduction. We believe the PoI framework is generalizable, but extending it to accommodate a broader class of evaluation metrics, particularly complex, non-differentiable, or sequence-level metrics related to safety and alignment, is a key avenue for future research.

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

# Appendix

## A LITERATURE REVIEW

In what follows, we present a more thorough literature review of the state of the art of verifiable compute and distributed training in the context of decentralized training of large-scale machine learning models.

### A.1 CRYPTOGRAPHIC PROOFS (ZKML)

One approach to trustless verification is to use cryptographic proofs, notably zero-knowledge succinct arguments (zk-SNARKs and STARKs Lavin et al. (2024)), to prove that training computations were carried out correctly. In principle, zk-SNARKs can provide a succinct proof of a large computation (like an LLM training step) that anyone can quickly verify on-chain, with the proof size and verify time independent of the model's size Thaler et al. (2022). This gives cryptographic guarantees of *correctness* – a malicious trainer cannot cheat the proof Thaler et al. (2022). In practice, however, compiling a massive neural network training into a SNARK circuit is extremely expensive, with recent analyses suggesting an increase of several orders of magnitude in overhead in computation cost and latency for even just inference tasks in a ZK circuit Chen et al. (2024). A recent framework (VerifBFL Bellachia et al. (2025)) demonstrated verifiable federated learning by generating zk-SNARK proofs for each participant's local training. While the results seem somewhat promising, they are still far from implementable. Indeed, for a relatively simple convolutional neural network trained on the MNIST handwritten dataset LeCun (1998), the authors observed that the on-chain verification was fast (¡0.6 s), but producing a proof for even a tiny training round took on the order of 81 seconds Bellachia et al. (2025). This overhead is prohibitive for large models or many training iterations. Fully zk-proving the training of a 70B-parameter LLM is impractical for the foreseeable future, absent breakthroughs in proof efficiency.

### A.2 PROOF-OF-LEARNING (POL)

Introduced by Jia et al. (2021), PoL leverages the fact that training a model (via, e.g., stochastic gradient descent or ADAM Kingma & Ba (2014)) produces a unique trajectory of model updates that is hard to forge without doing the work. In a PoL scheme, the prover (trainer) logs a sequence of intermediate states – e.g., model weights and hyper-parameters after each batch or epoch – along with metadata like batch indices and random seeds. This sequence is the "certificate" of training. A verifier can then randomly spot-check some of these intermediate steps: they pick a random subset of steps and re-compute the training transition (e.g., take the recorded weights at step $k$, apply the claimed gradient on the stated batch $k$, and check if it indeed produces the recorded weights at step $k+1$) Jia et al. (2021). If all checked steps are consistent, the verifier gains confidence that the entire sequence (from initial weights to final model) results from legitimate training. By adjusting how many steps are verified, one can trade off verification cost for assurance level. The security argument for PoL is that constructing a fake training log is as hard as training the model – essentially, inverting or short-cutting SGD is difficult. For example, an attacker would have to find gradients that produce a desired final model without actually computing them, which, in general, is computationally as expensive as honest training. A main drawback of PoL is its computational cost. Indeed, it is shown in Jia et al. (2021) that the time complexity of the verification step evolves as

$$\mathsf{Cost}(\texttt{PoL}) = \mathcal{O}(E \cdot Q \cdot \ell \cdot C_{|\theta|}), \tag{16}$$

where $E \in \mathbb{N}$ is the number of epochs in the training algorithm, $Q \in \mathbb{N}$ is the number of verifications per epoch, $\ell \in \mathbb{N}$ is the so-called checkpointing interval (i.e., how often the protocol checkpoints) and $C_{|\theta|} \in \mathbb{R}_+$ is the computational cost associated with training a model as a function of its weights, $\theta \in \mathbb{R}^d$, which is notably large for LLMs. Notice that Equation equation 16 only considers a single verifier, which can, in turn, lend itself to collusion. In a more general setting, one would employ a committee of $M$ verifiers but would need to increase the computational cost in equation 16 by a factor of $M$, accordingly.

Furthermore, the computational cost of PoL induces a *Verifier's Dilemma*: verifying many steps can be costly, so if not adequately incentivized, verifiers might be lazy and skip checks, undermining security.

In order to accommodate for potential discrepancies arising from, e.g., differences in floating point algebra, Jia et al. (2021) proposes to take proof as valid if the output weights from the provider and the validator $\theta_{\text{end}}^{\text{prover}}, \theta_{\text{end}}^{\text{verifier}}$, respectively, are sufficiently close. More precisely, given some measure of distance $d : \mathbb{R}^d \times \mathbb{R}^d \to \mathbb{R}_+$ and some tolerance $\mathsf{Tol} > 0$, a proof is taken as valid in their setting if $d(\theta_{\text{end}}^{\text{prover}}, \theta_{\text{end}}^{\text{verifier}}) < \mathsf{Tol}$.

Recent extensions of PoL incorporate stronger incentive models. For example, in Zhao et al. (2024), the authors propose a "capture-the-flag" game where verifiers earn extra rewards by finding any inconsistency (flags) in the proof, ensuring they check diligently. We intend to explore, improve, and expand these techniques and extend them to create (i) more computationally efficient methodologies and (ii) a base protocol with fully distributed training.

Another variant proposed in Zhang & Wang (2024) is the so-called *Proof-of-Sampling Protocol (PoSP)*. In their model, computations are taken as valid with probability $1 - p$ and otherwise verified by a committee of $M$ verifiers with probability $p$. Should a computation be deemed as *invalid*, the prover gets penalized an amount that is large enough so that, rationally, their best strategy is always to submit a *valid* computation. Put differently, the expected reward from *cheating* is smaller than the expected rewards from performing the computation. Intuitively, this approach incurs a cost of the order of

$$\mathsf{Cost}(\textsc{PoSP}) = \mathcal{O}\left(p \cdot M \cdot E \cdot Q \cdot \ell \cdot C_{|\theta|}\right), \tag{17}$$

which is an improvement over equation 16 provided that $pM < 1$, i.e., if the verification probability satisfies $p < 1/M$. This, in turn, creates a delicate balance. On the one hand, if one cares about fault tolerance, then $M$ must be relatively high, which means that the proportion of verified computations is small. This could, in turn, lead to ill behaviors from the provers or arbitrarily large potential penalties (often expressed as staked amounts), which might result in entry barriers from the protocol. These high entry barriers also imply that only a few providers and verifiers can join the network, which might, in turn, lead to centralization. On the other hand, reducing the size of the committee to its minimum (e.g., $M = 2$) would yield minimal computational gains while at the same time exposing the protocol to verifier collusion (intuitively, the smaller the verifying committee is, the easier it will be to manipulate).

### A.3 ROBUST AGGREGATION AND BYZANTINE-RESILIENT TRAINING

A related but distinct line of work addresses Byzantine-resilient distributed learning through *robust aggregation* methods. FLTrust Cao et al. (2021) proposes trust bootstrapping where a central server maintains a small "root" dataset and uses cosine similarity to weight client updates, filtering out malicious contributions. Other robust aggregators include coordinate-wise median, trimmed mean Yin et al. (2018), Krum Blanchard et al. (2017), and geometric median approaches.

**Distinction from our work.** Robust aggregation and PoI address complementary concerns:

- *Robust aggregation* filters contributions based on *similarity* to trusted data or other clients. It does not verify that workers actually performed training (they could submit scaled honest gradients or random noise that passes similarity checks).
- *PoI* certifies that the *outcome* (model improvement on a committed benchmark) was achieved, regardless of how individual updates were produced. It catches lazy workers who free-ride as well as adversaries whose attacks degrade evaluation metrics.

These mechanisms are orthogonal: one could deploy robust aggregation for Byzantine resilience at each round *and* PoI for end-to-end outcome verification. Our framework's process audits additionally verify that declared computations were actually performed, which robust aggregation alone cannot provide.

A.4    KNOWN VULNERABILITIES OF PROOF-OF-LEARNING

Fang et al. Fang et al. (2023) demonstrated attacks against the original PoL scheme Jia et al. (2021), showing that adversaries can forge training trajectories by exploiting the tolerance mechanism and by finding "shortcut" paths through weight space. Their attacks rely on:

1. *Tolerance exploitation*: Accumulating small per-step errors within $\tau$ that compound over many steps.
2. *Trajectory shortcuts*: Finding alternative sequences of updates that reach similar final weights without honest training.

Our protocol addresses these vulnerabilities through several mechanisms:

- *Tight tolerance calibration*: We calibrate $\tau$ to the p99 of honest replay drift under the declared stack $\Xi$, leaving minimal room for adversarial accumulation while preserving reproducibility.
- *Outcome verification (PoI)*: Even if an adversary forges a process-valid trajectory, PoI catches them if the final model fails to achieve claimed improvement on the committed evaluation set.
- *Economic deterrence*: The stake/slashing mechanism makes forgery attempts economically irrational even before detection, as expected losses exceed potential gains.
- *Commitment binding*: Our Merkle-based per-step commitments bind both input and output states, making trajectory shortcuts detectable if any intermediate state diverges.

The key insight is that PoI provides a "backstop": attacks that evade process verification must still produce a model that genuinely improved, which defeats the purpose of cheating.

A.5    DECENTRALIZED DISTRIBUTED TRAINING

Training large language models is often an exceedingly expensive computational task that requires computation due to their vast parameter sizes and data-intensive workloads. One common way of alleviating these computational costs is through distributing the computational load. While there is a vast literature on the topic, see, e.g., Sergeev & Del Balso (2018); Shoeybi et al. (2019); Huang et al. (2019); Rajbhandari et al. (2020); Lin et al. (2018) we will focus specifically on methods that allow for distributed training across multiple different machines in different locations. Central to this are the works of Douillard et al., Douillard et al. (2023; 2025) have proposed *Distributed Low-Communication* (DiLoCo) Douillard et al. (2023), a distributed optimization algorithm aimed at drastically reducing communication frequency in LLM training. Instead of synchronizing gradients at every minibatch, DiLoCo performs many local updates on each worker (using ADAM Kingma & Ba (2014) as the local optimizer) before occasionally averaging models across workers, using an outer loop with, e.g., *Nesterov momentum* Lin et al. (2019); Douillard et al. (2023). This approach allows training on "islands" of devices that are only intermittently connected, relaxing the typical requirement of a high-speed interconnect. DiLoCo achieved model quality on par with conventional data-parallel training on a standard large-scale dataset while communicating 500× less frequently among workers Douillard et al. (2023). In practical terms, eight workers communicating only once every 500 training steps matched the accuracy of fully synchronous training, demonstrating that vast reductions in communication are possible without sacrificing convergence. Moreover, DiLoCo was robust to heterogeneous data distributions across workers and resilient to dynamic availability of resources (workers can drop out or join during training with minimal impact) Douillard et al. (2023). Building on this idea, Douillard et al. (2025) introduced an enhanced strategy often referred to as Streaming DiLoCo, aiming to minimize communication overhead and latency penalties further Douillard et al. (2025). Streaming DiLoCo improves upon the original method by (i) partially synchronizing parameters, significantly reducing the peak bandwidth required at any given time Douillard et al. (2025), (ii) increasing efficiency in the implementation, and (iii) quantizing exchanged model updates to lower precision, cutting down the total volume of data transferred between workers Douillard et al. (2025). By combining these techniques, the authors were able to show that it is possible to distribute training of a billion-parameter transformer and reach similar accuracy as fully synchronous training while reducing required inter-worker bandwidth by about two orders of magnitude (a 100× reduction) Douillard et al. (2025). These low-communication approaches are significant because they enable multi-cluster or geographically distributed training of LLMs without the necessity of dedicated super-computing infrastructure.

# B  PROTOCOL SPECIFICATION (FULL DETAILS)

**Scope.** This appendix provides the complete protocol specification omitted from the main text for space: (i) single-prover Commit–Sample–Reveal (CSR), (ii) distributed CSR with always-on background worker audits and conditional escalation, and (iii) operational details on randomness, sampling, commitments, witnesses, and privacy.

## B.1  PUBLIC RANDOMNESS, SAMPLING, AND TIMING

**Window seed and timing.** For window $W$, all per-step commitments are frozen before sampling. A public randomness beacon (or VRF seed from SC) emits a window seed $r_W$ *after* commits freeze.

**Sampling.** Using $r_W$, we perform two VRF-based draws: (1) *workers* at background rate $\beta$ (distributed case), and (2) *per-worker steps* at rate $\alpha$. Draws are publicly verifiable and *without replacement* within a window; worker draws may also be without replacement over a rolling cycle to ensure coverage (main Eq. equation 3). Domain separation derives distinct seeds for worker vs. step draws (e.g., $\mathsf{H}(r_W \,\|\, \mathsf{workers})$, $\mathsf{H}(r_W \,\|\, \mathsf{steps} \,\|\, u)$).

## B.2  COMMITMENTS, WITNESS SCHEMA, AND PRIVACY

**Parameter sharding and Merkle roots.** Let model parameters be partitioned into fixed-size shards; let $C_{t-1} = \mathsf{MerkleRoot}(\theta_{t-1})$, $C_t = \mathsf{MerkleRoot}(\theta_t)$. A step commitment is the constant-size hash

$$h_t \;=\; H\big(C_{t-1} \,\|\, C_t \,\|\, W_t\big),$$

where the *witness* $W_t = (I_t, \Pi_t, O_{t-1}, R_t, \Lambda_t, \Xi)$ includes batch indices and Merkle proofs $(I_t, \Pi_t)$, the Merkle root of the optimizer state (or just the relevant shards) $O_{t-1}$, RNG seeds $R_t$, hyperparameters $\Lambda_t$, and the declared stack hash $\Xi$.

**Privacy model.** For audited steps, *committee-only* channels receive the required parameter/optimizer shards, metadata, seeds, and inclusion proofs. Public artifacts are per-step commitments and Merkle proofs. This preserves verifiability while limiting leakage. (LoRA/QLoRA reduces updated fraction $\kappa$; stateless optimizers reduce optimizer reveals.)

## B.3  SINGLE-PROVER CSR (FULL PSEUDOCODE)

---
**Algorithm 1** CSR (single-prover) per window $W$ with post-commit sampling

---
**Require:** Window size $G$; step audit rate $\alpha$; committee size $m$; tolerance
**Ensure:** Per-window verdict; audit logs
 1: **function** VRF_SAMPLE($U, p$; seed)
 2:   **return** a subset $S \subseteq U$ drawn by VRF with public seed; sample *without replacement*
 3: **end function**
 4: **function** VERIFYSTEP($t, \Xi,$)
 5:   Check (param-shards, opt-shards, metadata, seed) against Merkle roots $(C_{t-1}, C_t)$
 6:   Deterministically replay step $t$ on stack $\Xi$; compute discrepancy and compare to
 7:   **return** `pass`/`fail` + audit record (hashes, discrepancy)
 8: **end function**
 9: **Commit:** For $t = 1..G$, post $h_t$ binding $(C_{t-1}, C_t, W_t)$ to the ledger
10: **Freeze:** Close the window; all $h_t$ immutable
11: Draw public seed $r_W$
12: **Sample steps:** $S \leftarrow$ VRF_SAMPLE($\{1..G\}, \alpha; r_W$)
13: **for** $t \in S$ **in parallel do**
14:   Committee of size $m$ requests witness *to committee only*
15:   Members run VERIFYSTEP($t, \Xi,$); vote by supermajority ($\geq \lceil m/2 \rceil + 1$)
16:   **if** vote is `fail` **then**
17:     Slash; record audit and mark window `fail`
18:   **end if**
19: **end for**
20: **Finalize window:** Publish verdicts; apply slashing/penalties; proceed to next window

---

B.4 DISTRIBUTED CSR WITH BACKGROUND AUDITS AND ESCALATION (FULL PSEUDOCODE)

---

**Algorithm 2** Distributed CSR with Background Worker Sampling and Escalation (per window)

---

**Require:** Window size $G$; per-worker step audit rate $\alpha$; background worker sampling rate $\beta$; committee size $m$; tolerance ; escalation $(\alpha', \beta', m') \succeq (\alpha, \beta, m)$
**Ensure:** Per-window verdict; slashing decisions; audit logs
 1: **function** VRF_SAMPLE($U, p$; seed)
 2:     **return** a subset $S \subseteq U$ drawn by verifiable randomness (public seed), *without replacement*
 3: **end function**
 4: **function** VERIFYSTEP($u, t, \Xi,$)
 5:     Check witness schema (param-shards, opt-shards, metadata, seed) vs. Merkle root $C_{u,t}$
 6:     Deterministically replay step $t$ for worker $u$ on $\Xi$; compare discrepancy to
 7:     **return** `pass`/`fail` + audit record (hashes, discrepancy)
 8: **end function**
 9: **Stage 1 (always): Aggregation verification**
10: Recompute aggregator output from committed worker updates; verify aggregation commitments and indices
11: Record `AggregationPass/Fail`
12: Draw public randomness $r_W$ *after* all worker commits are finalized
13: **Stage 2a (always): Background worker audits**
14: $W_{\mathrm{bg}} \leftarrow$ VRF_SAMPLE(workers, $\beta$; $r_W$)
15: **for** worker $u \in W_{\mathrm{bg}}$ **do**
16:     $S_u \leftarrow$ VRF_SAMPLE(steps of $u$, $\alpha \cdot G$; $(r_W, u)$)
17:     **for** step $t \in S_u$ **in parallel do**
18:         Committee ($m$) requests witnesses (committee-only)
19:         Members run VERIFYSTEP($u, t, \Xi,$ ); vote; aggregate by supermajority
20:     **end for**
21:     **if** *any* step vote is `fail` **then**
22:         Slash $u$; record attribution $\langle u, S_u \rangle$; mark window `fail`
23:     **end if**
24: **end for**
25: **Stage 2b (conditional): Escalation on failures**
26: **if** (`AggregationFail`) **OR** (`PoIFail at milestone`) **then**
27:     $(\alpha, \beta, m) \leftarrow (\alpha', \beta', m')$                            $\triangleright \beta' = 1$ allowed
28:     Draw new randomness $r'_W$
29:     $W_{\mathrm{esc}} \leftarrow$ VRF_SAMPLE(workers, $\beta'$; $r'_W$)
30:     **for** worker $u \in W_{\mathrm{esc}}$ **do**
31:         $S'_u \leftarrow$ VRF_SAMPLE(steps of $u$, $\alpha' \cdot G$; $(r'_W, u)$)
32:         **for** step $t \in S'_u$ **in parallel do**
33:             Committee re-verifies with $m'$; slash on `fail`
34:         **end for**
35:     **end for**
36:     */* Optionally apply escalation to implicated preceding windows for attribution. */*
37: **end if**
38: **Finalize window:** Publish verdicts and hash-only logs; apply slashing; proceed to next window

---

B.5 PROTOCOL TIMELINE

Figure 2 shows the temporal structure of the windowed CSR protocol. Key timing guarantees:

- *Post-commit randomness*: Window seed $r_W$ is drawn *after* all step commits are finalized, preventing adaptive step selection.
- *Pipelined execution*: While window $W$ is audited, the prover may commit steps for window $W+1$; latency contributes $O(1)$ per window, not per step.
- *Deadlines*: Reveals must arrive within $\Delta_{\mathrm{reveal}}$; votes within $\Delta_{\mathrm{vote}}$. Missed deadlines trigger penalties.

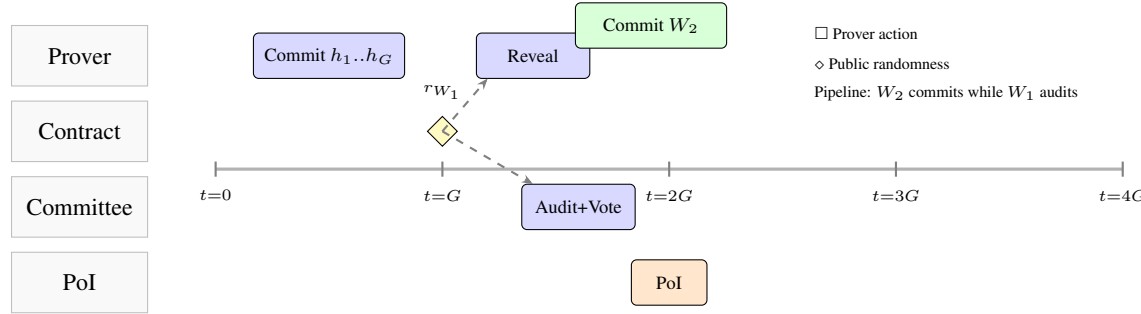

Figure 2: **Protocol timeline** showing pipelined execution. Commits for window $W_2$ (green) overlap with audit of window $W_1$. Public randomness $r_W$ is drawn post-commit to prevent adaptive attacks. PoI milestones run in parallel with training audits.

## C PROOFS

**Lemma 1** (Base Law for Single Forgery) Let $\alpha \in (0, 1]$ be the fraction of audited steps and $q \in (0, 1]$ be the probability that an honest-majority committee correctly identifies a forgery. For a single forged step ($f = 1$), the detection probability is $\delta(1) = \alpha q$.

*Proof.* Let $A$ be the event that the single forged step is sampled for an audit, and let $B$ be the event that the verifier committee correctly detects the forgery. The sampling is uniform and random, thus $P(A) = \alpha$. The conditional probability of detection, given the step is sampled, is defined as $P(B|A) = q$. Since the sampling event and the committee's verification are independent, the total probability of detection is the joint probability:

$$\delta(1) = P(A \cap B) = P(B|A)P(A) = q\alpha$$

$\square$

**Remark (committee quality factor).** In the analysis we decompose $q$ as

$$q = \underbrace{1 - P_{\text{maj-Byz}}(M, F, m)}_{\text{honest majority}} \cdot \underbrace{1 - P_{\tau\text{-miss}}}_{\text{no masking by } \tau},$$

cf. Eq. equation 6. For deterministic replay, $\tau=0$ so $P_{\tau\text{-miss}}=0$.

**Consequence (multi-step baseline).** For $f \geq 1$ forgeries within a window and sampling without replacement, the per-worker detection obeys

$$\delta(f) \geq 1 - (1 - \alpha q)^f,$$

with equality under independent draws

**Theorem 1** (Efficient Frontier for Process Audits) Let $\ell \in \mathbb{N}_+$ be the total number of training steps, with per-step costs $C_p$ for the Prover and $C_v$ for a Verifier. Let $m \in \mathbb{N}_+$ be the committee size. For a target single-step detection probability $\delta^* \in (0, q]$, the minimum verification cost is given by:

$$\text{Cost}_{\min}(\delta^*; m) = \ell C_p + \frac{\delta^*}{q}\ell m C_v$$

*Proof.* The total expected cost is the sum of the Prover's computation cost and the expected verification cost:

$$\text{Cost}_{\text{total}}(\alpha, m) = \ell C_p + \mathbb{E}[\text{Verification Cost}] = \ell C_p + (\alpha\ell)m C_v$$

To achieve a target soundness level $\delta^*$, we require $\delta(1) = \delta^*$. By Lemma 1, $\alpha q = \delta^*$, which implies the necessary sampling fraction is $\alpha = \delta^*/q$. Since $\alpha \leq 1$, it must hold that $\delta^* \leq q$. Substituting this expression for $\alpha$ into the cost function yields the minimal cost for the target soundness $\delta^*$:

$$\text{Cost}_{\min}(\delta^*; m) = \ell C_p + \left(\frac{\delta^*}{q}\right)\ell m C_v$$

This establishes a linear trade-off between the verification cost and the soundness guarantee. $\square$

**Design note.** Targets with $\delta^* > q$ are infeasible without improving $q$ (e.g., increasing $m$ or tightening $\tau$). Integer constraints enter only through $\alpha \ell \in \mathbb{N}$; rounding $\alpha$ up preserves the bound.

**Theorem 2** (Pipelined Liveness) Let $\ell \in \mathbb{N}_+$ be the total number of steps, processed in windows of size $G \in \mathbb{N}_+$. Let $T_{upd}$ be the per-step computation time, $\Delta$ be the network delay, and $\Delta_{aud}$ be the audit finalization time. The total execution time is bounded by:

$$T_{\text{total}} \leq \ell T_{upd} + \left\lceil \frac{\ell}{G} \right\rceil (2\Delta + \Delta_{aud}) + O(1)$$

*Proof.* The total time $T_{\text{total}}$ is the sum of the Prover's sequential computation time and the cumulative latency from the audit pipeline. The total computation time across all $l$ steps is $\ell \cdot T_{upd}$. The protocol is processed in $N_w = \lceil \ell/G \rceil$ windows. The pipelined design ensures that the latency of auditing window $w$ overlaps with the computation of window $w+1$. Thus, latency contributes an overhead per window, not per step. The latency for one window consists of at least two network delays (commit finalization, reveal) and the audit time $\Delta_{aud}$. The total cumulative latency is $N_w(2\Delta + \Delta_{aud})$. Combining these terms, we obtain the upper bound on the total time. □

**Operational clarification.** In our implementation, the reveal and vote phases have explicit deadlines ($\Delta_{\text{reveal}}, \Delta_{\text{vote}}$); missed deadlines are slashable and keep the $O(1)$ per-window contribution. Stage 2a (background) and Stage 2b (escalation) run within the same pipeline and hence do not add per-step latency.

**Theorem 3** (Economic Security via Staking) Let $\mathcal{G} > 0$ be the Prover's gain from a single successful cheat and let $s_p > 0$ be the Prover's stake. Honesty is a strictly dominant strategy if:

$$s_p > \left( \frac{1}{\alpha q} - 1 \right) \mathcal{G}$$

*Proof.* Let the utility of honesty be $U_{\text{honest}} = 0$. The expected utility of attempting a single cheat, $\mathbb{E}[U_{\text{cheat}}]$, is determined by the two possible outcomes: success (no detection) or failure (detection). The probability of detection is $\delta(1) = \alpha q$.

$$\mathbb{E}[U_{\text{cheat}}] = P(\text{success}) \cdot (\text{Gain}) + P(\text{failure}) \cdot (\text{Loss})$$

$$\mathbb{E}[U_{\text{cheat}}] = (1 - \delta(1)) \cdot \mathcal{G} + \delta(1) \cdot (-s_p)$$

For honesty to be strictly dominant, we require $\mathbb{E}[U_{\text{cheat}}] < 0$:

$$(1 - \delta(1))\mathcal{G} - \delta(1)s_p < 0 \implies (1 - \delta(1))\mathcal{G} < \delta(1)s_p$$

Solving for the stake $s_p$ yields:

$$s_p > \frac{1 - \delta(1)}{\delta(1)} \mathcal{G}$$

Substituting $\delta(1) = \alpha q$ from Lemma 1 gives the required condition:

$$s_p > \frac{1 - \alpha q}{\alpha q} \mathcal{G} = \left( \frac{1}{\alpha q} - 1 \right) G$$

This ensures that the expected utility of cheating is negative, making it an economically irrational strategy. □

**Corollary 3.1 (background sampling; multiple forgeries).** For a worker forging $f$ steps in a window with background audits at rate $\beta$, the per-window detection is

$$\delta_{\text{bg}}(f) = \beta \left[ 1 - (1 - \alpha q)^f \right] \quad \text{(cf. Eq. equation 2)}.$$

Let $\mathcal{G}(f)$ be the per-window cheating gain. Then honesty is strictly dominant if

$$s_p > \frac{1 - \delta_{\text{bg}}(f)}{\delta_{\text{bg}}(f)} \mathcal{G}(f),$$

equivalently $(1 - \delta_{\text{bg}})G_\$ - \delta_{\text{bg}}D < 0$ in the reward/deposit view.

**Lemma 2** (Few-Step Influence Bound). We prove that adversarial perturbations on a subset of training steps have bounded cumulative impact.

*Proof.* Consider the parameter trajectory under honest execution $\{\theta_t^{\mathrm{hon}}\}_{t=0}^T$ and adversarial execution $\{\theta_t^{\mathrm{adv}}\}_{t=0}^T$, starting from the same $\theta_0$. Let $\mathcal{F} \subseteq \{1, \ldots, T\}$ be the set of forged steps with $|\mathcal{F}| = f$.

For $t \notin \mathcal{F}$ (honest steps), both trajectories apply the same update:

$$\theta_t^{\mathrm{adv}} - \theta_t^{\mathrm{hon}} = \theta_{t-1}^{\mathrm{adv}} - \theta_{t-1}^{\mathrm{hon}} - \eta(\nabla\mathcal{L}(\theta_{t-1}^{\mathrm{adv}}) - \nabla\mathcal{L}(\theta_{t-1}^{\mathrm{hon}})).$$

By $L$-smoothness, $\|\nabla\mathcal{L}(\theta_{t-1}^{\mathrm{adv}}) - \nabla\mathcal{L}(\theta_{t-1}^{\mathrm{hon}})\| \leq L\|\theta_{t-1}^{\mathrm{adv}} - \theta_{t-1}^{\mathrm{hon}}\|$, so:

$$\|\theta_t^{\mathrm{adv}} - \theta_t^{\mathrm{hon}}\| \leq (1 + \eta L)\|\theta_{t-1}^{\mathrm{adv}} - \theta_{t-1}^{\mathrm{hon}}\|.$$

For $t \in \mathcal{F}$ (forged steps), the adversary introduces deviation $\Delta_t$:

$$\|\theta_t^{\mathrm{adv}} - \theta_t^{\mathrm{hon}}\| \leq (1 + \eta L)\|\theta_{t-1}^{\mathrm{adv}} - \theta_{t-1}^{\mathrm{hon}}\| + \|\Delta_t\|.$$

Unrolling the recursion from $t = 0$ (where the deviation is zero) to $t = T$:

$$\|\theta_T^{\mathrm{adv}} - \theta_T^{\mathrm{hon}}\| \leq \sum_{t \in \mathcal{F}} (1 + \eta L)^{T-t}\|\Delta_t\|.$$

With gradient clipping at norm $B$, each forged gradient is bounded: $\|\Delta_t\| \leq 2\eta B$ (the difference between the clipped adversarial and honest gradients). The loss deviation follows from $L$-smoothness:

$$|\mathcal{L}(\theta_T^{\mathrm{adv}}) - \mathcal{L}(\theta_T^{\mathrm{hon}})| \leq L\|\theta_T^{\mathrm{adv}} - \theta_T^{\mathrm{hon}}\| + \frac{L}{2}\|\theta_T^{\mathrm{adv}} - \theta_T^{\mathrm{hon}}\|^2.$$

For small deviations (typical regime), the quadratic term dominates. $\square$

**Remark (Adam and momentum optimizers).** For Adam, the analysis extends by noting that the adaptive learning rate is bounded: with $\epsilon > 0$ and $\beta_2 < 1$, we have $\eta_{\mathrm{eff}} \leq \eta/\sqrt{\epsilon}$. Momentum introduces exponentially decaying memory, which can amplify deviations by at most a factor of $1/(1 - \beta_1)$. The qualitative bound $O(f \cdot \eta B \cdot \mathrm{poly}(T))$ remains.

## C.1 PoI Sample Size Guidance

Table 5 provides guidance for selecting the number of evaluation samples $n$ (or blocks $n_{\mathrm{eff}}$) to achieve target statistical power under different correlation structures.

| Correlation structure | Effective $n$ | Target power $\geq 0.90$ | Target power $\geq 0.99$ |
|---|---|---|---|
| i.i.d. tokens | $n$ | $n \geq 50$ | $n \geq 100$ |
| Block correlation ($\rho \approx 0.3$) | $n_{\mathrm{eff}} \approx n/2$ | $n \geq 100$ | $n \geq 200$ |
| Strong block correlation ($\rho \approx 0.6$) | $n_{\mathrm{eff}} \approx n/4$ | $n \geq 200$ | $n \geq 400$ |
| Sequence-level metrics (BLEU, safety) | $n_{\mathrm{seq}}$ | $n_{\mathrm{seq}} \geq 200$ | $n_{\mathrm{seq}} \geq 500$ |

Table 5: Sample size guidance for PoI under different correlation structures. Values assume a moderate effect size ($\gamma/\sigma \approx 0.5$) and one-sided $t$-test at $\alpha_{\mathrm{stat}} = 0.05$. For smaller effect sizes, scale $n$ by $(\sigma/\gamma)^2$.

**Effective sample size under block correlation.** When evaluation tokens within a block of size $b$ have correlation $\rho$, the variance of the block mean is inflated by a factor $(1 + (b-1)\rho)$ relative to i.i.d. sampling. The effective sample size for $n$ total tokens in $n/b$ blocks is:

$$n_{\mathrm{eff}} = \frac{n}{1 + (b-1)\rho}.$$

For blocked sampling (recommended), we draw entire blocks via VRF and compute paired differences at the block level, achieving the effective sample size of $n/b$ independent observations.

# D  COST AND BANDWIDTH DETAILS

This appendix formalizes prover/verifier overheads and network reveals in closed form, and derives deployable reductions that correspond to the summary equations in the main paper.

## D.1  NOTATION AND REGIMES

Let $P \in \mathbb{N}_+$ denote the number of trainable parameters and $u > 0$ the bytes per parameter (e.g., $u=2$ for bf16/fp16). Define the parameter byte size

$$P := u\,P.$$

Let $\varphi \geq 0$ denote the optimizer state multiplier (Adam: $\varphi=2$ for $(m, v)$; stateless SGD: $\varphi=0$). Let $\kappa \in (0, 1]$ be the effective fraction of parameters updated and therefore revealed by an audited step (full-parameter training: $\kappa=1$; LoRA/QLoRA: $\kappa \ll 1$). Fix a shard size $S > 0$ (bytes) and write

$$L_p := \left\lceil \frac{P}{S} \right\rceil, \qquad L_o := \varphi\,L_p,$$

for the numbers of parameter and optimizer shards, respectively. Let $\pi > 0$ denote the mean Merkle inclusion proof length (bytes) per shard. A training window contains $G \in \mathbb{N}_+$ steps, with audited step fraction $\alpha \in (0, 1]$ and background worker sampling rate $\beta \in (0, 1]$. Committees have size $m \in \mathbb{N}_+$ and vote under tolerance $\tau \geq 0$; the committee correctness factor is $q = q(m, \tau) \in (0, 1]$. Let $T_{\text{step}} > 0$ be the prover wall-time per training step and $C_v > 0$ the verifier replay time per audited step (on stack $\Xi$). The effective uplink bandwidth is $\text{BW} > 0$. A pipeline overlap coefficient $\chi \in [0, 1]$ models how much network transfer is hidden by compute ($\chi = 0$ fully hidden; $\chi = 1$ no overlap). For distribution to committees, define a dissemination factor $\delta_{\text{dist}} \in [1, m]$ ($\delta_{\text{dist}}=1$ for content-addressed/multicast; $\delta_{\text{dist}}=m$ for naive unicast).

## D.2  REVEAL BYTES PER AUDITED STEP

For one audited step, the prover reveals parameter and, if applicable, optimizer shards sufficient for deterministic replay, together with Merkle inclusion proofs. A precise and implementation-agnostic approximation is

$$B_{\text{step}} \approx \underbrace{\kappa\,P}_{\text{parameters}} + \underbrace{\kappa\,\varphi\,P}_{\text{optimizer}} + \underbrace{\kappa\,(L_p + L_o)\,\pi}_{\text{proofs}} = \kappa\,(1 + \varphi)\,(P + L_p\,\pi), \tag{18}$$

using $L_o = \varphi L_p$. Thus, relative to full-parameter training with Adam ($\kappa=1, \varphi=2$), LoRA/QLoRA scales bytes proportionally to $\kappa \ll 1$, and switching to stateless SGD ($\varphi=0$) removes the optimizer term.

**Merkle proof bound.**  For a $b$-ary Merkle tree with digest size $d$ bytes and $L$ leaves, a standard path proof satisfies

$$\pi \leq (b - 1)\,d\,\lceil \log_b L \rceil, \tag{19}$$

up to small framing constants. In our accounting, $L$ is on the order of $L_p$ (parameters) or $L_o$ (optimizer).

## D.3  PER-WINDOW NETWORK TRAFFIC

Let $B_{\text{step}}$ be given by equation 18. If a sampled worker reveals $\alpha G$ steps in a window, then the prover's *uplink* bytes are

$$B_{\text{win}}^{\uparrow} = \alpha\,G\,\delta_{\text{dist}}\,B_{\text{step}}, \tag{20}$$

where $\delta_{\text{dist}}$ captures distribution mode (1 for content-addressed/multicast; $m$ for naive unicast to a committee of size $m$).

### D.4 COMMIT/SERIALIZATION TIME AND INCREMENTAL MERKLE ROOTS

Let $T_{\text{commit}} > 0$ be the time per step to hash shards and update Merkle roots, and let $T_{\text{ser}} > 0$ be the time to serialize witnesses. We report the following normalized fractions,

$$\text{commit\_frac} := \frac{\text{median}(T_{\text{commit}})}{T_{\text{step}}}, \qquad \text{serialize\_frac} := \frac{\text{median}(T_{\text{ser}})}{T_{\text{step}}}. \qquad (21)$$

A naive implementation recomputes the full Merkle tree each step; a practical implementation updates the root in $O(\log L_p)$ per modified shard via cached internal nodes. Streaming shard hashing during the optimizer update overlaps a large fraction of $T_{\text{commit}}$ with backpropagation and reduces equation 21.

### D.5 END-TO-END NORMALIZED OVERHEAD

Normalizing to the prover's training time, the end-to-end overhead as a function of $\alpha$ satisfies

$$\rho(\alpha) \approx \underbrace{\frac{\alpha \, C_v}{T_{\text{step}}}}_{\text{verifier replay}} + \underbrace{\frac{T_{\text{commit}}}{T_{\text{step}}}}_{\text{prover commit}} + \underbrace{\chi \, \alpha \, \frac{\delta_{\text{dist}} \, B_{\text{step}}}{\text{BW} \, T_{\text{step}}}}_{\text{network}}, \qquad (22)$$

which complements the cost–soundness frontier of the analysis: the first term scales linearly in $\alpha$, the second is the no-audit baseline, and the third depends on the network regime and dissemination mode.

### D.6 MEASUREMENT PROTOCOL (REPRODUCIBLE)

Fix a stack $\Xi$ and dataset/configuration as in the experiments. Define four measured quantities:

$$T_{\text{commit}}, \quad T_{\text{ser}}, \quad B_{\text{step}}, \quad C_v.$$

Obtain $T_{\text{commit}}$ and $T_{\text{ser}}$ by instrumenting per-step hashing and serialization; report median and p95 and normalize by $T_{\text{step}}$ via equation 21. Obtain $B_{\text{step}}$ by instantiating equation 18 for (i) full-parameter+Adam and (ii) LoRA/QLoRA with/without optimizer moments, sweeping shard size $S$ and inserting $\pi$ from equation 19 (or empirical proofs). Obtain $C_v$ by replaying audited steps on $\Xi$. Finally, evaluate $\rho(\alpha)$ from equation 22 for $\alpha \in \{0.005, 0.01, 0.02, 0.05\}$, $\chi \in \{0, 0.5, 1\}$, and $\text{BW} \in \{100 \text{ Mb/s}, 1 \text{ Gb/s}\}$.

### D.7 DEPLOYABLE REDUCTIONS (CLOSED-FORM EFFECTS)

Let $B_{\text{step}}^{(X)}$ denote the step-bytes under intervention $X$.

**Stateless/low-state optimizers.** Setting $\varphi \to 0$ (SGD or quantized moments) replaces

$$B_{\text{step}} = \kappa(1 + \varphi)(P + L_p\pi) \quad \longmapsto \quad B_{\text{step}}^{(\text{SGD})} = \kappa(P + L_p\pi),$$

a multiplicative reduction by the factor $\frac{1}{1+\varphi}$ (i.e., $\approx 1/3$ vs Adam with $\varphi = 2$).

**Checkpoint-based verification.** Revealing full optimizer state every $k \in \mathbb{N}_+$ audited steps yields the bound

$$B_{\text{step}}^{(k)} \leq \kappa\left(1 + \frac{\varphi}{k}\right)(P + L_p\pi),$$

so the optimizer contribution shrinks by the factor $\frac{1+\varphi/k}{1+\varphi}$ (approaching the stateless limit as $k \to \infty$). Verifier compute rises modestly to replay from the nearest revealed checkpoint.

**Content-addressed serving.** Replacing naive unicast by content-addressed/multicast changes $\delta_{\text{dist}}$ in equation 20–equation 22 from $m$ to 1, dividing network egress by $\approx m$ and lowering the network term of $\rho(\alpha)$ accordingly.

**Streaming Merkle roots.** Incremental, pipelined hashing reduces $T_{\text{commit}}$ and hence the second term in equation 22, without changing $B_{\text{step}}$.

**Lossless/fixed-point encodings.** If witnesses are encoded with lossless compression or fixed-point formats that preserve the commitment (hashing over the encoded bytes), then for some $\eta \in (0, 1)$ one has

$$B_{\text{step}}^{(\text{enc})} = \eta \, B_{\text{step}},$$

which linearly reduces the network term of equation 22.

### D.8 LoRA/QLoRA RULE-OF-THUMB

Let $\kappa_{\text{LoRA}} \ll 1$ denote the adapter-to-model parameter ratio. With Adam on adapters,

$$B_{\text{step}} \propto \kappa_{\text{LoRA}} \, (1 + \varphi) \, P \quad \text{(dominant param+moment bytes)},$$

while with stateless SGD on adapters,

$$B_{\text{step}} \propto \kappa_{\text{LoRA}} \, P.$$

Hence reveal bytes and commit overhead both drop by a factor $\approx 1/\kappa_{\text{LoRA}}$ relative to full-parameter training and by an additional factor $\approx 1/(1 + \varphi)$ when moments are removed.

### D.9 SANITY BOUNDS AND REGIMES

Under full-parameter training with Adam ($\kappa{=}1, \varphi{=}2$) and a binary Merkle tree ($b{=}2$) with digest size $d$,

$$B_{\text{step}} \lesssim 3 \, P + 3 \, L_p \, \pi, \qquad \pi \leq d \left\lceil \log_2 L_p \right\rceil,$$

so $B_{\text{step}}$ is at most a small constant multiple of the model size and scales sublinearly with $L_p$ via equation 19. In compute-bound regimes where

$$\chi \, \alpha \, \frac{\delta_{\text{dist}} \, B_{\text{step}}}{\text{BW}} \ll T_{\text{step}},$$

the network term of equation 22 is negligible—consistent with our microbenchmarks.

# E    COMMITTEE SIZING AND TOLERANCE CALIBRATION

This appendix provides exact and asymptotic formulas for the committee correctness factor

$$q(m,\tau) \;=\; \big(1 - P_{\text{maj-Byz}}(M, F, m)\big) \cdot \big(1 - P_{\tau\text{-miss}}(\tau, \Xi)\big),$$

together with closed-form sizing rules for $m$ at a given capture level $F/M$ and calibration of the numerical tolerance $\tau$ under a declared execution stack $\Xi$.

## E.1    COMMITTEE CAPTURE MODEL AND EXACT MAJORITY RISK

Let $M \in \mathbb{N}_+$ be the total verifier population and $F \in \{0, \dots, M\}$ the number of Byzantine verifiers (capture). When a committee of size $m \in \{1, \dots, M\}$ is drawn uniformly without replacement, the number of Byzantine verifiers in the committee,

$$X \;\sim\; \text{Hypergeom}(M, F, m), \qquad \mathbb{P}[X = x] \;=\; \frac{\binom{F}{x}\binom{M-F}{m-x}}{\binom{M}{m}}, \quad x = 0, \dots, m.$$

Write $s_{\text{maj}} := \lceil m/2 \rceil$ for the *majority* threshold and $s_{\text{sup}} := \lceil m/2 \rceil + 1$ for the *strict supermajority* threshold used by the voting rule. The probability that a committee is (at least) Byzantine-majority is

$$P_{\text{maj-Byz}}(M, F, m) \;=\; \mathbb{P}[X \geq s_{\text{maj}}] \;=\; \sum_{x=s_{\text{maj}}}^{m} \frac{\binom{F}{x}\binom{M-F}{m-x}}{\binom{M}{m}}. \tag{23}$$

Accordingly, the *honest-majority* factor is $1 - P_{\text{maj-Byz}}(M, F, m)$, which appears multiplicatively in $q(m, \tau)$.

**Binomial approximation and KL tail.**    When $m \ll M$, sampling without replacement is well approximated by $X' \sim \text{Binomial}(m, \rho)$ with $\rho := F/M$. Using a Chernoff–Cramér bound,

$$\mathbb{P}[X' \geq s_{\text{maj}}] \;\leq\; \exp\Big(- m\, D\big(\tfrac{1}{2} \,\|\, \rho\big)\Big), \qquad D(p\|q) := p\ln\frac{p}{q} + (1-p)\ln\frac{1-p}{1-q}. \tag{24}$$

Thus a sufficient condition for $P_{\text{maj-Byz}} \leq \varepsilon$ is

$$m \;\geq\; \frac{\ln(1/\varepsilon)}{D\big(\tfrac{1}{2} \,\|\, \rho\big)} \quad \text{(odd $m$ rounded up)}, \tag{25}$$

with a finite-population correction improving equation 25 by the factor $(M - m)/(M - 1)$ when $m$ is not negligible relative to $M$.

## E.2    TOLERANCE CALIBRATION AND NUMERIC-MISS PROBABILITY

For a given audited step, let $\widehat{\theta}_t$ be the recomputed state under honest replay on stack $\Xi$, and consider a norm $\| \cdot \|_\mathsf{X}$ used by the committee. Define the *honest drift* random variable

$$E_{\text{hon}} \;:=\; \big\|\widehat{\theta}_t - \theta_t\big\|_\mathsf{X} \quad \text{under honest execution across replicas of $\Xi$}.$$

Fix a target false-positive rate $\eta \in (0, 1)$ for honest steps. The tolerance $\tau$ is chosen as the $(1 - \eta)$-quantile of the honest drift,

$$\tau \;:=\; F_{E_{\text{hon}}}^{-1}(1 - \eta) \quad \Longrightarrow \quad \mathbb{P}\big[E_{\text{hon}} \leq \tau\big] \;\geq\; 1 - \eta. \tag{26}$$

Let $E_{\text{adv}}$ denote the discrepancy under a forged update (after deterministic replay of the forged step). The probability that numeric tolerance masks a true error is

$$P_{\tau\text{-miss}}(\tau, \Xi) \;:=\; \mathbb{P}\big[E_{\text{adv}} \leq \tau\big]. \tag{27}$$

When the adversary induces a deviation of magnitude at least $\delta_{\min}$ in the chosen norm, and $E_{\text{hon}}$ is stochastically dominated by a sub-Gaussian proxy with variance proxy $\sigma^2$, a simple bound follows from Gaussian tails:

$$P_{\tau\text{-miss}}(\tau, \Xi) \;\leq\; \Phi\Big(\frac{\tau - \delta_{\min}}{\sigma}\Big) \;\overset{\tau = F_{E_{\text{hon}}}^{-1}(1-\eta)}{\lesssim}\; \Phi\Big(\Phi^{-1}(1 - \eta) - \frac{\delta_{\min}}{\sigma}\Big), \tag{28}$$

where $\Phi$ is the standard normal CDF. In practice we estimate $F_{E_{\text{hon}}}$ empirically on $\Xi$ and validate $P_{\tau\text{-miss}}$ via fault-injection sweeps.

### E.3 PUTTING IT TOGETHER: HITTING A TARGET $q^\star$

For a target correctness $q^\star \in (0, 1)$, one may choose $(m, \tau)$ by solving

$$\left(1 - P_{\text{maj-Byz}}(M, F, m)\right) \cdot \left(1 - P_{\tau\text{-miss}}(\tau, \Xi)\right) \geq q^\star. \tag{29}$$

A constructive sizing is:

(i) pick $\varepsilon_{\text{maj}}, \varepsilon_\tau > 0$ with $(1 - \varepsilon_{\text{maj}})(1 - \varepsilon_\tau) \geq q^\star$;   (ii) choose $m$ by equation 23 or equation 25 so that $P_{\text{maj-Byz}} \leq \varepsilon_{\text{maj}}$;

When $\tau = 0$ (deterministic replay), $P_{\tau\text{-miss}} = 0$ and equation 29 reduces to the majority condition.

### E.4 MINIMAL ODD COMMITTEE SIZES (EXACT HYPERGEOMETRIC)

Table 6 reports the minimal odd $m$ achieving $1 - P_{\text{maj-Byz}}(M, F, m) \geq q_{\text{target}}$ for representative capture fractions $\rho = F/M$ (computed from equation 23 in the limit $M \to \infty$ with fixed $\rho$, i.e., binomial tails; values agree with the finite-$M$ hypergeometric for $M \gg m$).

| $q_{\text{target}}$ | $\rho = 0.05$ | 0.10 | 0.20 | 0.30 | 0.40 |
|---|---|---|---|---|---|
| $\geq 0.99$ | 3 | 5 | 11 | — | — |
| $\geq 0.95$ | 3 | 3 | 7 | 15 | — |

Table 6: Minimal odd $m$ such that $\mathbb{P}[\text{honest majority}] \geq q_{\text{target}}$ for selected capture fractions $\rho = F/M$. Dashes indicate infeasibility under majority voting at that $q_{\text{target}}$.

# F  PROTOCOL DIAGRAMS

## F.1  SINGLE-PROVER PoL

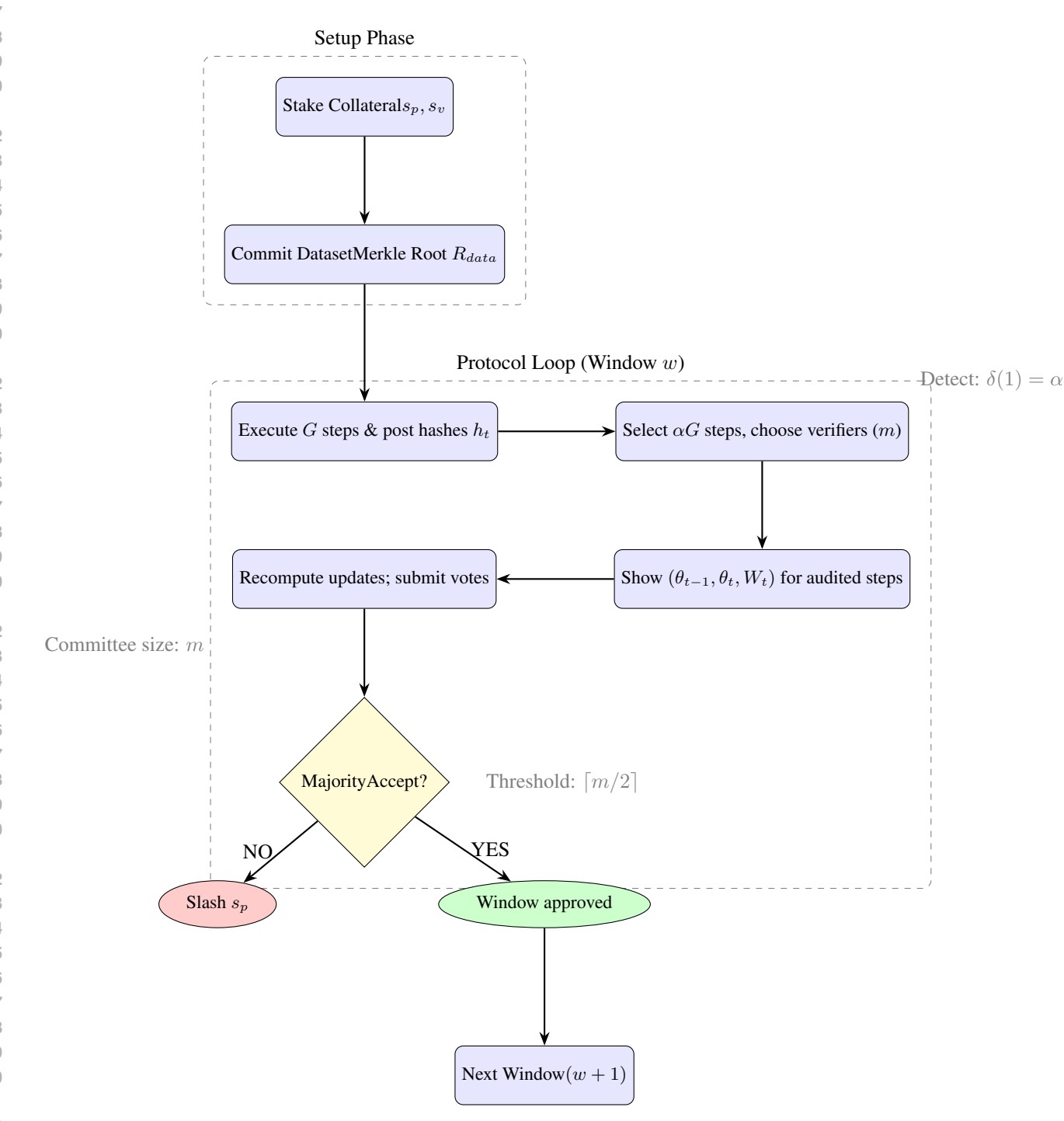

Figure 3: Proof of Learning Protocol: Single prover

## F.2 DISTRIBUTED PoL

Figure 4: Verifiable DiLoCo. Figure adapted from Douillard et al. (2023)

## G STATEMENT ON THE USE OF AI

In preparing this manuscript, we utilized large language models as a productivity tool. Their assistance was helpful for improving the clarity and tone of the writing, for grammatical and consistency checks, during initial research ideation, and for debugging segments of the experimental code. The final content and all intellectual contributions are the authors' own.

