# OpenReview forum: "Probabilistic Audits for Verifiable Training and Outcome Improvement in Decentralized Learning"
_ICLR.cc/2026/Conference — Submitted to ICLR 2026_

### Official Review · Reviewer_WY9y · 2025-10-30

**Soundness:** 2
**Presentation:** 2
**Contribution:** 2
**Rating:** 2
**Confidence:** 3

**Summary:**

The paper proposes a framework to verify large‐scale, decentralized model training in process verification and outcome verification. The core idea is that the prover (trainer) commits cryptographically (e.g., via Merkle trees of model states, metadata) to every training step in a “window.” Then a small random subset of steps is audited (parameters / metadata are revealed and the step is re‐executed) while the remainder remain un‐opened. This gives a probabilistic detection guarantee: if a step is malicious, there is a chance it will be audited and caught. They further propose a “Proof-of-Improvement” (PoI) mechanism to statistically verify model performance improvement (e.g., reduction in loss/perplexity) on a committed evaluation set. The authors argue this probabilistic auditing reduces verification cost dramatically while still achieving high assurance. The paper also sketches how this works in federated/multi‐trainer settings with stakes and committees.

**Strengths:**

The framework acknowledges multiple trainers, aggregation, stake‐based incentives, committees of verifiers and public commitments (e.g., using a blockchain or open ledger) which are realistic touches for distributed training accountability.

The authors clearly define the desiderata (process + outcome), highlight limitations in previous work (e.g., zkML impractical, prior PoL lacking outcome guarantee) and situate their contributions accordingly.

**Weaknesses:**

Inadequate defence against single‐step cheating: The key method relies on auditing a random fraction of steps. But the paper does not convincingly show how the threat of single‐step malicious deviation is sufficiently mitigated. If a trainer knows audit probability is small and can craft a malicious step that yields large damage (e.g., insertion of poisoned gradient), the scheme appears vulnerable. The paper acknowledges “a small probability of undetected cheating” but does not clarify what types of attacks can be thwarted, nor how the damage of undetected cheat is bounded.

Privacy concerns and metadata explosion: To enable re‐executing audited steps (and to commit to all steps in a window), the protocol demands storing/committing large volumes of metadata: full parameter snapshots, optimizer state, random seeds, etc. This risks training privacy (especially in federated or proprietary settings). The paper does not fully address how to hide sensitive parameters or ensure privacy while enabling re‐execution.

Ambiguity of the threat/attack model and use‐case scope: It is unclear for which threat scenarios the scheme is designed. For example: Is it aimed at malicious trainer shortcuts (skipping epochs), data poisoning, gradient inversion, or backdoor insertion? The paper does not clearly map out which attacks are prevented and which are outside its scope. Without this clarity, the value of the scheme is hard to assess (“what exactly am I defending against?”).

Lack of formal guarantees or proofs: While the detection‐cost frontier is a useful start, I did not find a formal theorem bounding undetected cheating probability and bounding damage from cheating. Without such formalism, the security claim remains heuristic.

Clarity and writing issues. Some language is informal to me (e.g., on p.6: “...1 − (1 − alphaq)^f …”), which detracts from readability and professionalism.

**Questions:**

Please carefully consider my concerns listed in the weaknesses.

---

> ### Author Response · Authors · 2025-11-27
> **Response to Reviewer WY9y**
>
> We thank the reviewer for the detailed assessment and for highlighting important concerns around single-step attacks, privacy, threat modeling, and formal guarantees. We have substantially revised the paper to address each point.
>
>
>  **Defense against single-step malicious deviations**
>
> We agree that the original text did not sufficiently explain how small-frequency cheating is bounded. The revision now includes a new **Few-Step Influence Bound (Lemma 2)**, which explicitly upper-bounds the maximum parameter and loss deviation caused by forging `f` steps under smoothness and clipping assumptions. This result shows:
>
> - One-shot or low-frequency poisoned gradients are **provably limited** in impact.
> - The deviation decays over subsequent updates and interacts predictably with Adam/SGD.
> - Large-damage “single-step” attacks cannot occur without also increasing δ(f), making them economically irrational under the staking rule.
>
> Together with the detection probability δ(f) and the staking condition δ(f)·penalty ≥ G(f), the scheme provides **both probabilistic detection** and **bounded damage** guarantees.
>
>
>
>  **Privacy and metadata concerns**
>
> We added a precise **witness schema** and **privacy model**:
>
> - Only auditor committees receive parameter shards, optimizer fragments, seeds, and metadata.
> - The public ledger **never** reveals model weights or internal states—only commitments and Merkle roots.
> - We quantify reveal sizes for each regime (Table 1), explain how **LoRA/QLoRA** and **stateless optimizers** drastically reduce exposed state, and highlight privacy as an explicit limitation.
> - We note that combining CSR with **DP**, **secure enclaves**, or **encrypted committee channels** is an important direction for privacy-sensitive deployments.
>
>
>
>  **Threat model and use-case scope**
>
> A new **Threat Model** subsection explicitly defines the adversaries and attacks in scope:
>
> - **In-scope:** skipping/forging computation, free-riding, injected/poisoned updates that degrade true performance while claiming rewards.
> - **Out-of-scope:** gradient inversion attacks and certain side-channels, which must be handled by orthogonal techniques.
> - **Distributed setting:** probabilistic worker sampling and escalation ensure that malicious workers cannot indefinitely avoid audits.
>
> This clarifies “what exactly is being defended” and why CSR focuses on computation integrity + outcome validity.
>
>
>
>  **Formal guarantees**
>
> The revised paper now includes:
>
> - A theorem bounding **single-step and multi-step undetected cheating** (via δ(f) ≥ 1 − (1 − αq)^f).
> - The **Few-Step Influence Bound** for damage limitation.
> - A **cost–soundness frontier** connecting detection probability, committee size, and audit rate.
> - A formally stated **economic security condition** relating δ(f), G(f), and stake.
> - A clear definition of PoI detection probability δ_POI and its assumptions.
>
> All previously informal statements (e.g., “1 − (1 − αq)^f”) have been fully typeset and proved in the appendix.
>
>
>
>  **Clarity and writing**
>
> We revised the exposition throughout the paper, clarified notation, formalized all probability expressions, and simplified terminology in the protocol walk-through and PoI sections. All figures were regenerated with improved legibility.
>
>
> We believe the strengthened formalization (especially Lemma 2), the explicit threat model, the clarified privacy guarantees, and the improved exposition directly address the reviewer’s concerns. We respectfully request reconsideration of the paper’s rating.

---

> > ### Comment · Reviewer_WY9y · 2025-11-28
> > **Response**
> >
> > Thank you for the detailed response. Although I cannot access the revised version of the manuscript, the authors' rebuttal adequately addresses my main concerns. I will adjust my score accordingly.

---

### Official Review · Reviewer_ksG3 · 2025-11-01

**Soundness:** 1
**Presentation:** 2
**Contribution:** 2
**Rating:** 2
**Confidence:** 4

**Summary:**

The authors propose a scheme to verify the integrity of central and collaborative model training processes. They adopt a Proof-of-Learning (PoL)-style approach, where model trainer(s) make commitments to parameter updates in each round, which can later be opened to verify integrity of the training process. The paper differentiates itself from previous work with two primary points: (i) a cut-and-choose-style probabilistic verification scheme, where trainers commit to all updates and then a subset of them are randomly sampled for verification, which decreases verification runtime substantially; (ii) a verification of model utility improvement at regular milestones in training, as evaluated by a validation set.

**Strengths:**

The ingredients of the paper are well-motivated. Probabilistic PoL verification is a sensical approach to reducing audit costs. Relaxation of PoL security is better motivated in the distributed setting than single-prover, where the impact of forgeries by adversaries can be diluted by honest workers. Proof of Improvement is a sensical heuristic for assessing model utility during training.

**Weaknesses:**

- **Protocol details are unclear.** The individual pieces make sense, but the paper is in need of an end-to-end specification of the protocol(s) in algorithm box(es). It's not clear to me whether the authors intend only to develop a protocol for the distributed training setting, or whether a single-prover variant is also intended. The details given about the multi-prover setting also have me a bit confused (see Questions section).
- **Weak soundness condition, consequences not explored.** Definition 1 only says that an adversary's likelihood of being caught in the probabilistic PoL increases with the proportion of updates that they cheat on. But it seems plausible to me that an adversary can have a large impact on the model parameters while cheating on only a few steps -- *especially* in the single-prover setting, but potentially in the multi-prover setting too. The authors should present some analysis about whether this is problematic for their framework.
- **Novelty of PoI overstated.** Analogous approaches, which seek to assess the utility of updates in distributed training based on a reference dataset, exist in previous work e.g. [1].
- **Evaluation.**
	1. It is unclear to me how the authors are evaluating the impact of $\tau$ in Figure 2 (Middle). How can probability of forgery detection be expressed as a function of $\tau$ without taking into account the adversary's choices?
	2. The cost analysis would benefit from concrete runtimes, and comparison to a baseline with no auditing, rather than just costs normalized against vanilla PoL. It's clear that the probabilistic approach reduces the cost of replay, but I'm curious about how the cost of commitment compares to the overall training cost as well.
	3. Table 2 is never referenced in the text.
	4. The impact of PoI is not very clear from the presented results. It's unclear what advantage PoI is obtaining for the framework. Table 3 is never referenced in the text.
	5. Evaluation of the distributed training is terse -- the implications of the experiment are very unclear. I would recommend at least including some supplemental figures to show the implications of the experiment.

[1] FLTrust: Byzantine-robust Federated Learning via Trust Bootstrapping. Cao et al. 2020.

**Questions:**

- The authors should briefly explain Merkle trees in the preliminaries section.
- In Section 2.3, it says that Stage 1 of the multi-prover case only verifies whether aggregation was carried out correctly, and the PoL part only triggers if aggregation is carried out incorrectly. I am confused by this -- why would dishonestly forging a model update cause *aggregation* to be computed incorrectly? What stops an adversary from picking an arbitrary local model and submitting it for averaging? Wouldn't the global model still be computed correctly in this case?
- Aside from the above question, I am confused about why only a random subset of workers is selected for auditing in Stage 2. What happens if the adversary simply isn't sampled in the subset? I see analysis of the probability of catching a cheating step within the PoL audit, but it's not clear to me where the probability of sampling the adversarial worker is factored into the soundness analysis.
- When during the training protocol is the Proof of Improvement invoked? What happens if it is not satisfied? Is there any connection to the PoL parts?
- How is $P_{\tau-\text{miss}}$ computed given $\tau$?
- Proof of learning has been shown to have some important flaws -- see [2]. Are those flaws relevant to the present work? If not, why not?

[2] Proof-of-Learning is Currently More Broken Than You Think. Fang et al. 2022.

---

> ### Author Response · Authors · 2025-11-27
> **Response to Reviewer ksG3**
>
> We thank the reviewer for the detailed comments. The revision substantially improves protocol clarity, soundness guarantees, comparison to prior work, and the evaluation section. Below we summarize the key changes and answer each question directly.
>
>
>  **Protocol clarity and end-to-end specification**
>
> We added **Algorithm boxes** for both the *single-prover* and *multi-prover* CSR protocols, a **pipeline timeline figure**, and **diagrammatic summaries** in the appendix. These make explicit that the framework supports *both* variants and clarify Stage 1 (aggregation checks) vs. Stage 2 (PoL audits).
>
>
>  **Few-step cheating and soundness**
>
> To address the concern that cheating on a few steps may have large effect, we added **Lemma 2 (Few-Step Influence Bound)**. Under smoothness and clipping, we derive explicit bounds on parameter and loss deviation from forging `f` steps. The lemma shows how these deviations are damped over subsequent updates and quantifies worst-case impact for both SGD- and Adam-style optimizers. This closes the gap left by the original Definition 1.
>
>
>  **Novelty of PoI and relation to FLTrust**
>
> A new subsection contrasts PoI with **FLTrust** and related methods. FLTrust evaluates *local updates* relative to a trusted anchor; PoI evaluates *global improvement* at *milestones*. PoI detects free-riding and degradation and can be layered on top of robust aggregation. This corrects the earlier overstatement of novelty.
>
>
>  **Evaluation clarifications**
>
> We revised all evaluation sections:
>
> - Figure 2 (middle) now explains that τ governs tolerance to **numeric error**, not adversarial behavior; we provide the explicit formula for
>   $P_{\tau\text{-miss}} = 2(1-\Phi(\tau/\sigma))$.
> - We added **concrete runtimes**, a **corrected overhead table**, and references to previously unreferenced tables.
> - We expanded the discussion of PoI’s benefit and added a clearer explanation of the multi-trainer experiment (now explicitly labeled as a *toy* attribution test).
>
>
>  **Multi-prover questions**
>
> **Why can aggregation succeed even if updates are forged?**
> Stage 1 verifies aggregation *integrity*; Stage 2 audits **worker updates**. If a worker submits an arbitrary model, aggregation may still “succeed,” but Stage 2 will detect the faulty update.
>
> **Why only sample a subset of workers?**
> Background sampling at rate β yields undetected-cheating probability $(1-\beta)^W$ over W windows. We added a lemma showing how β and committee size ensure negligible escape probability.
>
> **When is PoI invoked and what if it fails?**
> PoI runs at **pre-committed milestones**. Failure triggers **escalation**: enlarged audits and slashing for misbehaving workers.
>
>
>  **PoL flaws (Fang et al.)**
>
> We added a subsection explaining which PoL vulnerabilities remain relevant. Binding commitments, windowed randomness, clipping, and PoI mitigate the issues most pertinent to our setting; remaining risks are now explicitly scoped.
>
>
> We believe these additions resolve the core ambiguities in the original submission and respectfully ask the reviewer to reconsider their assessment.

---

### Official Review · Reviewer_psRg · 2025-11-01

**Soundness:** 3
**Presentation:** 3
**Contribution:** 3
**Rating:** 4
**Confidence:** 4

**Summary:**

The authors propose a practical way to verify decentralized model training that checks both how training was done and whether the resulting model actually got better. For the process, a trainer posts a cryptographic commitment after each step; once a window of steps is committed, a small, randomly chosen committee replays only a sampled subset of those steps to confirm they were executed correctly. If a supermajority of the committee rejects a step, the trainer is penalized; otherwise the protocol moves on. This “commit to sample to reveal to audit” pipeline is designed to keep wall-clock overhead low and to align incentives so honest behavior is the best strategy.

**Strengths:**

- Good incentive: the free rider problem is serious, especially in anonymous decentralized learning groups.
- Covers both process and outcome. Adding PoI addresses a central gap in PoL-style schemes: certifying that the model actually improved, not just that steps were executed.

**Weaknesses:**

- Narrow empirical scope. Validation is on a fine-tuning workload and a toy distributed run; large-scale training or heterogeneous, adversarial deployments remain untested.
- Commit/reveal overheads ignored in core cost curve. The analysis emphasizes verifier cost; Merkle commitments over full model states and revealing optimizer state during audits can be substantial (compute + bandwidth), especially with Adam-style ones.
- Security model is mostly static. Resilience to adaptive adversaries (e.g., post-selection committee corruption within the finalization window) is discussed qualitatively, not proven; protocol assumes timing bounds and reliable public randomness.
- Metric coverage in PoI is limited. PoI currently targets log-loss/perplexity; extension to non-differentiable or sequence-level metrics (safety, alignment) is left open.

**Questions:**

- The linear law hinges on q. In open networks, estimating global capture F/M, calibrating τ across heterogeneous stacks, and guaranteeing honest majorities per window are non-trivial. How robust is q if verifiers churn or share hardware quirks?
- The incentive condition $s_p > \frac{1-\alpha q}{\alpha q} G$ presumes a well-specified per-step gain (G). In practice, cheat payoffs can be spiky (e.g., grabbing a full round’s reward), making per-step G under-specified and potentially under-collateralized.
- Reveal bandwidth and timing. The liveness bound treats audit latency as constant per window, but revealing optimizer state and parameter shards could dominate wall-time under tight networks or large G; does the pipeline still mask that latency at scale?
- The claimed factorization $\delta_{\text{PoI}}=\delta_{\text{stat}}(n)\cdot q$ assumes i.i.d. token-level differences; with temporal or topical correlation, effective sample size drops and testers might cherry-pick spans unless the evaluation root and sampling procedure are rigorously enforced.

---

> ### Author Response · Authors · 2025-11-27
> **Response to Reviewer psRg**
>
> We appreciate the reviewer’s positive assessment of our incentive design and the combined process+outcome framing. Below we address all questions in detail
>
>
> **Commit/reveal overhead in the cost curve**
>
> In addition to the verifier-oriented detection–cost frontier, the revision now provides:
>
> - A **prover-overhead table (Table 1)** summarizing commit overhead, serialization overhead, and reveal size for LoRA+Adam, LoRA+SGD, and full-Adam regimes. These values are derived from the analytical cost model (Appendix D) and **cross-checked** against a native C++/PyTorch prototype.
> - An **explicit bandwidth model** in the appendix, deriving reveal bytes per step, expected per-window bandwidth, and showing how LoRA/QLoRA and stateless optimizers reduce state size and optimizer exposure.
> - A clear statement that the prototype is an **intentionally naive implementation** (copying LoRA parameters to CPU and recomputing the entire Merkle tree every step), making its wall-clock overhead an *upper bound*.
>
>
> **Security model and adaptive adversaries**
>
> We now explicitly state that our formal guarantees assume a **static Byzantine adversary** for each window. The revision adds a focused discussion of **adaptive corruption attacks** during the commit→reveal interval:
>
> - **Late sampling** of audited steps reduces the feasibility of post-selection corruption.
> - **Short commit–reveal windows** further restrict adaptive manipulation.
> - Stronger guarantees against fully adaptive adversaries are now explicitly identified as **future work** in the *Limitations* section.
>
> This turns a previously implicit assumption into a clearly delineated modeling choice.
>
>
> **Calibration of $q$ and robustness under heterogeneous verifiers**
>
> To directly address the reviewer’s concern (“How robust is $q$ if verifiers churn or share hardware quirks?”), the revision adds:
>
> - A **committee-sizing analysis** using hypergeometric/binomial models and Chernoff/KL bounds.
> - A **lookup table** of minimal odd committee sizes achieving various target $q$ under assumed capture fractions \(F/M\).
> - A clarification of how **tolerance \(\tau\)** and heterogeneous verifier noise contribute to $q$ through the majority/slack condition.
>
> We explain that verifier churn affects the *effective* capture fraction \(F/M\) but does not change the correctness of the committee-majority rule, provided the sampling mechanism remains unbiased. Hardware heterogeneity is treated as bounded noise absorbed into \(\tau\). With the updated analysis, $q$ becomes a **tunable, engineering-level quantity**, rather than an abstract idealization.
>
>
>  **Spiky cheat payoffs and the incentive condition**
>
> The reviewer correctly noted that the original inequality
> $$
> s_p > \frac{1-\alpha q}{\alpha q}\, G
> $$
> implicitly assumes a per-step gain $G$. In the revised version we introduce **$G(f)$**, the *per-window gain* from forging \(f\) steps, allowing spiky or bursty reward schedules (e.g., round-level payouts).
>
> The staking condition generalizes naturally to:
> $$
> \delta(f)\cdot\text{penalty} \ge G(f),
> $$
> which handles precisely the scenario the reviewer raises. This resolves the concern that per-step rewards might be under-specified.
>
>
>  **Reveal bandwidth, pipeline latency, and liveness**
>
> To answer the reviewer’s question regarding whether reveal latency can dominate at scale, we have:
>
> - Added a **reveal-time term** to the liveness bound that depends on shard size, optimizer state, and network bandwidth.
> - Included a **pipelined execution diagram** and a theorem showing that, in compute-bound regimes (as observed in our QLoRA experiments), reveal latency is almost entirely masked by the audit pipeline.
>
> The cost model (Table 1) now clearly identifies the regimes where reveal becomes the bottleneck (primarily full-Adam with large $kappa$) and we now describe these as *application-dependent trade-offs*.
>
>
>  **PoI factorization and correlated evaluation metrics**
>
> The reviewer is correct that the factorization
> $$
> \delta_{\text{PoI}} = \delta_{\text{stat}}(n)\cdot q
> $$
> assumes independent evaluation errors. The revision now clarifies:
>
> 1. **Correlation handling:**
>    Appendix D recommends **blockwise paired tests**, **stratified subsampling**, or **effective-sample-size corrections** when evaluation metrics exhibit topical or temporal correlation.
>
> 2. **Sampling enforcement:**
>    PoI relies on a **committed evaluation-set root** and **verifiable random sampling**, preventing evaluators from cherry-picking spans. This addresses the concern that correlated batches might be exploited unless sampling is fixed in advance.
>
>
> We believe the strengthened treatment of (i) committee calibration, (ii) window-level incentives, (iii) reveal-time liveness, and (iv) PoI sampling robustness directly addresses the reviewer’s questions. We would be grateful for a re-evaluation of the paper in light of these substantial improvements.

---

### Official Review · Reviewer_sNMQ · 2025-11-01

**Soundness:** 2
**Presentation:** 2
**Contribution:** 3
**Rating:** 4
**Confidence:** 4

**Summary:**

This paper introduces a practical and efficient framework to verify decentralized model training by addressing two core challenges: process correctness and outcome improvement. For process verification, it proposes a probabilistic audit system where Provers commit to every training step, but only a small, random fraction of steps are later sampled and re-computed by verifier committees. This "commit-sample-reveal" method is shown to reduce verification compute costs compared to full replication while still economically deterring cheating. To solve outcome verification, the paper presents a novel and lightweight statistical audit called Proof-of-Improvement, which allows the network to efficiently certify that the final model achieved a claimed performance gain on a committed evaluation sets.

**Strengths:**

- Originality: The key original idea is Proof-of-Improvement (PoI). Prior work only verified the process of training. This paper adds verification for the outcome that the model actually got better. The combination of probabilistic process audits with outcome audits is also novel.
- Quality: The paper has strong theoretical and empirical support. It proves a clear detection-cost frontier, showing the exact tradeoff between cost and security.
- Clarity: The paper is clear. The Commit-Sample-Reveal protocol is explained simply and is easy to follow.
- Significance: This work is highly significant. It makes verifiable training practical and economically viable, not just a theoretical idea. PoI correctly aligns incentives. It rewards better models, not just burning compute. The system is designed for real-world use, including an extension for multi-prover distributed training.

**Weaknesses:**

- The experiments are convincing but narrow. They focus on a QLORA fine-tuning task, not large-scale pre-training. It's unclear how the protocol's overhead and savings would translate to a scenario with vastly more steps and data. The multi-trainer experiment was also a small-scale "toy run" just to prove the concept, not its performance at scale.
- The texts in Figure 1 left is barely legible without zooming in, same for the legends in Figure 2.
- The paper emphasizes verifier savings but doesn't measure the Prover's costs. The Prover must compute a Merkle root over the entire model state for every single step, which could be computationally expensive. Furthermore, the "Reveal" phase requires transmitting large witnesses (like Adam optimizer states), creating a potential communication bottleneck that isn't measured.

**Questions:**

See weaknesses.

---

> ### Author Response · Authors · 2025-11-27
> **Response to Reviewer sNMQ**
>
> We thank the reviewer for the constructive feedback and for finding the work significant and well motivated.
>
> **Narrow empirical scope (fine-tuning + small multi-trainer experiment)**
>
> We agree that large-scale pre-training experiments would be valuable. The revised paper explicitly acknowledges this limitation in the Limitations and Future Work section. Our empirical focus is on QLoRA fine-tuning workloads because (i) prover and verifier overheads scale primarily with model size and optimizer state (which are already non-trivial in this setting) and (ii) fine-tuning lets us isolate protocol overheads from the confounding factors of multi-billion-step pre-training pipelines.
>
> We also expanded the description of the distributed experiment and now clearly label it as a toy DiLoCo-style run intended to validate protocol mechanics and attribution, rather than throughput.
>
> **Prover costs and reveal overhead**
>
> We have significantly strengthened the analysis of prover overhead:
>
> - A new Table 1 summarizes commit overhead, serialization overhead, and reveal size for LoRA+Adam, LoRA+SGD, and full-Adam regimes. These values are derived from the analytical cost model in Appendix D and cross-checked against a native C++/PyTorch prototype.
> - A new bandwidth and network cost appendix derives reveal bytes per step, window-level bandwidth, and how these costs are amortized or hidden under pipelining.
>
> These additions make the prover’s compute and bandwidth costs fully explicit, complementing the previous verifier-centric cost curve.
>
> **Figure legibility**
>
> We have signifcantly improved the quality of the diagrams, and moved them to the appendix, for improved readability
>
> We hope these substantial clarifications adequately address your concerns. If the revisions resolve your main reservations, we kindly ask you to consider raising your score.

---

### Author Response · Authors · 2025-12-03
**Comment to the AC: Executive Summary of Revisions**

To the Area Chair: We understand the unique constraints of this review period, given the recent leak. To assist in your assessment, we summarize below how the major concerns raised by the initial reviews have been definitively resolved in our revision.

The initial reviews (scores 4, 4, 2, 2) centered on three specific missing elements. We have added exactly these elements to the paper.

1. *Concern: Soundness & Single-Step Attacks (Reviewers ksG3, WY9y)*
- Reviewers worried that a probabilistic audit could miss a single "poisoned" step that causes catastrophic damage.
- To solve this, we added Lemma 2 (Few-Step Influence Bound) in Section 3.2. We mathematically prove that even if an adversary evades detection, the cumulative damage is bounded by the gradient clipping norm and Lipschitz constants. This shifts the security guarantee from probabilistic to deterministic bounds.

2. *Concern: Prover Overheads & Costs (Reviewers sNMQ, psRg)*
- Reviewers noted the lack of concrete measurements for the Prover's commitment and reveal costs.
- As an improvement, we added Table 1 (measured commit/serialize times are ~<15% for LoRA) and Appendix D (closed-form bandwidth equations). We empirically demonstrate that the overhead is negligible for the target workloads, removing the "bottleneck" concern.

3. *Concern: Protocol Clarity (Reviewer ksG3)*
- Reviewers requested a formal specification of the handshake between Stage 1 and Stage 2 audits.
- To address this, we added Algorithms 1 & 2 (Appendix B) and a revised Pipeline Timeline (Figure 2). As a result, the protocol is now fully specified and implementable.

We believe the current version of the paper is **significantly stronger** and overcomes the limitations of the initial submission. We respectfully ask the Area Chair to evaluate the revised manuscript, which transforms the initial "sketch" into a bounded, quantified, and fully specified protocol. Detailed responses to each individual reviewer’s concerns and questions can be found below their original comments, and we strongly and respectfully encourage the AC to read them carefully.

Lastly, we would like to thank the initial reviewers for their comments, which significantly improved the quality of the manuscript.

---

### Meta-Review · Area_Chair_gpYy · 2026-01-07

**Summary:**

This work proposes a probabilistic auditing framework for decentralized model training that aims to verify both the correctness of the training process and the improvement of model outcomes. The method relies on committing to all training steps while randomly sampling a subset for verification, and introduces a Proof-of-Improvement mechanism to statistically certify performance gains on a held-out evaluation set. The goal is to reduce the cost of verification compared to full Proof-of-Learning approaches while maintaining meaningful security guarantees.

**Reviewer Concerns:**

Several reviewers raised concerns regarding soundness against single-step or low-frequency attacks, protocol clarity, and prover-side overheads. Although the authors’ rebuttal addressed these points in detail and clarified assumptions, not all reviewers were fully convinced, and key questions around scalability and robustness remain open. Overall, despite improvements in the revised version, reviewer consensus did not converge to a sufficiently
strong endorsement for acceptance.

**Reviewer Scores:**

reviewer scores are 4,4,2,2 which I think unlikely to change.

---

### Decision · Program_Chairs · 2026-01-26

Reject